# OPTIMIZING THE INEFFABLE: GENERATIVE POLICY LEARNING FOR HUMAN-CENTERED DECISION-MAKING

## ABSTRACT

Algorithmic decision-making is widely adopted in high-stakes applications affecting our daily lives but often requires human decision-makers to exercise their discretion within the process to ensure alignment. Explicitly modeling human values and preferences is challenging when tacit knowledge is difficult to formalize, as Michael Polanyi observed, "We can know more than we can tell." To address this challenge, we propose generative near-optimal policy learning (`GenNOP`). Our framework leverages a conditional generative model to reliably produce diverse, near-optimal, and potentially high-dimensional stochastic policies. Our approach involves a re-weighting scheme for training generative models according to the estimated probability that each training sample is near-optimal. Under our framework, decision-making algorithms focus on a primary, measurable objective, while human decision-makers apply their tacit knowledge to evaluate the generated decisions, rather than developing explicit specifications for the ineffable, human-centered objective. Through extensive synthetic and real-world experiments, we demonstrate the effectiveness of our method.

## 1 INTRODUCTION

We have witnessed a pragmatic shift in automated decision-making systems from a heavily first-principles approach (*e.g.*, the DENDRAL (Buchanan & Feigenbaum, 1981), the MYCIN (Van Melle et al., 1984), and the INTERNIST-1 (Miller et al., 1986) expert systems from the 1960–80s) towards a mostly empirically-grounded one (*e.g.*, IBM Watson for Oncology (Strickland, 2019), automated insulin delivery (AID) systems (Sherr et al., 2022), and the COMPAS assessment for recidivism risk (Dressel & Farid, 2018) from the 2000–20s), as the availability of empirical data and the capacity to model it grow in orders of magnitude. The apparent success of data-driven systems is typically evidenced by their superior predictive accuracy and calibration on held-out evaluations. This aligns with classic results showing that statistical (actuarial) aggregation outperforms unaided clinical judgment for quantifiable information (Meehl, 1954; Dawes et al., 1989; Grove et al., 2000). Yet this success masks a crucial oversight, as decision-makers often conflate the *positive* capabilities of their empirical toolkits (what can be predicted or optimized) with the inherently *normative* nature of decision-making (what ought to be done). This conflation has produced many unintended consequences: decisions that are optimal in a statistical sense but misaligned with human values.

Consider a critical care physician attending to a patient just admitted to the intensive care unit (ICU). The physician can observe the conditions of the patient, gather her demographic information and medical history, and order a series of tests. Suppose the physician has access to a database of patient characteristics ($X$), actions taken by critical care physicians ($A$), and clinical outcomes ($Y$), as well as an algorithm ($f$) derived from this database that can give accurate and well-calibrated predictions of $Y$ given $X, A$. Should the physician simply adopt the solution to the optimization problem $\arg\max_a f(X = x, A = a)$?

The algorithm provides a positive statement: "patients with characteristics $x$ can expect a $y$-day reduction in length of stay at the ICU if $a$ dosage of medications is administered to them"; however, adopting the $\arg\max$ implies a normative statement: "reducing the length of stay is the sole objective of the patient's care". This implication holds for any one or combination of quantifiable

clinical outcomes. Instead, the critical care physician's true normative statement is: "we should treat Ms Wang with $a$ dosage of medications because we believe this is the best course of action for her care", stressing the importance of the individual (Tonelli, 1998). Any care derived from the true normative statement necessitates the clinical judgment by the physician to reflect the *unquantifiable* characteristics and welfare of the patient and to strike a balance among quantifiable clinical outcomes, the patient's agency for their own care, and medical ethics and best practices. We term the class of decision-making tasks where the importance of the individual requires human judgment *human-centered decision-making* problems.

To model the effect of human judgment and to formalize normative statements, we introduce two implicit quantities: (1) a human-centered objective $U$ that can be evaluated by human judgment but never measurable; and (2) an overall utility $V$ whose order determines true normative preferences but is *ineffable* to both humans and algorithms. Figure 1 illustrates the relationship among the measurable objective $Y$, the human-centered objective $U$, and the overall utility $V$, for a given action space. See Section 2 for more details. Several examples of human-centered decision-making problems in various domains and their corresponding $A, Y, U, V$ are included in Appendix B.

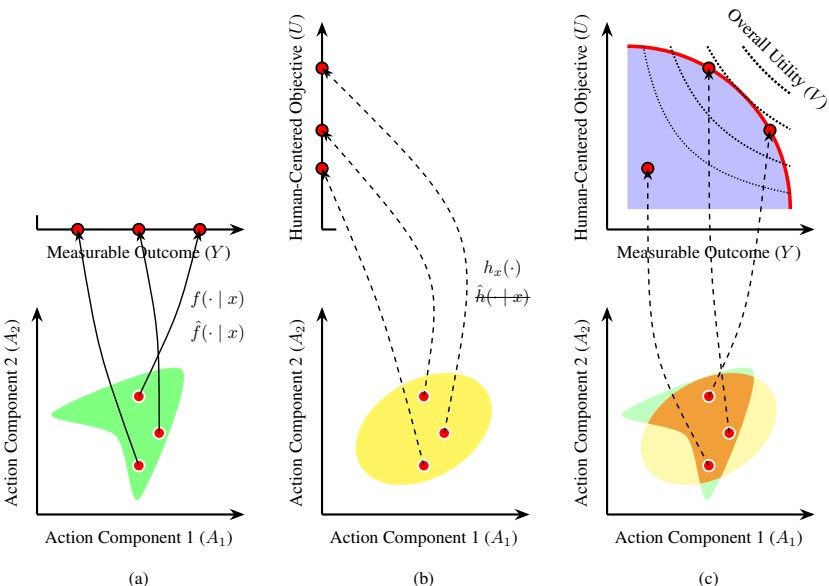

Figure 1: Illustration of the Objectives of Human-Centered Decision-Making Problems: (a) The measurable objective $Y$ is the quantity that can be measured and optimized by algorithms: we can know the relationship between $Y$ and $A$ as well as the feasible region of $A$ through explicit knowledge $f(\cdot \mid x)$ and/or learning the contextual mapping $\hat{f}(\cdot \mid x)$ from empirical data; (b) The human-centered objective $U$ is the quantity that can be evaluated by human judgment but never measurable: human evaluators, upon observing a number of actions, can determine the relative contextual preferences among the actions through their tacit knowledge $h_x(\cdot)$, which cannot be formalized as $\hat{h}(\cdot \mid x)$ and reapplied without the human evaluators involved; (c) We have effectively a bi-objective optimization problem with one objective ($Y$) accessible to algorithms and the other ($U$) inaccessible to algorithms but visible to human evaluators: trade-offs between the two objectives are inevitable, and the overall utility $V$ is a function of the two objectives.

A natural solution to injecting human values into automated decision-making systems is to place humans-in-the-loop (HITL). However, most current HITL systems either:

1. attempt to create a proxy for human judgment either by directly eliciting human tacit knowledge (Polanyi, 1966) or through preference-based learning such as reinforcement learning from human feedback (RLHF) (Ouyang et al., 2022). However, human judgment is too complex to capture comprehensively under all contexts, observable and latent; once human judgment is formalized, it can no longer adapt to individualities, distributional shifts, and

subtle changes in context, thereby losing the flexibility that made it valuable in the first place; or

2. reduce human involvement to simply accepting or rejecting singular algorithmic recommendations. An over-simplified human role can lead to both algorithmic aversion and over-reliance (Dietvorst et al., 2015; Banker & Khetani, 2019). When humans reject singular algorithmic recommendations, they resort to either making local perturbations to the recommended decisions or coming up with *de novo* decisions on their own, leaving valuable algorithmic power untapped.

Neither approach captures the strengths of both humans and algorithms. A more promising strategy is to design for complementarity (McLaughlin & Spiess, 2024; Hemmer et al., 2024): clearly delineating roles so that humans and algorithms each operate where they excel. We adopt such a strategy and propose a framework termed "generative near-optimal policy learning" (`GenNOP`). Under our framework: (1) Human experts define a measurable objective $Y$ and set $\epsilon$, the acceptable gap from optimal $Y$-value. (2) A generative model, trained on empirical $(X, A, Y)$ data, produces a distribution $\pi$ over actions that achieve at least $(1 - \epsilon)$ of the optimal $Y$-value. (3) Human experts sample candidate actions from $\pi$ and select the one that maximizes their judgment of $U$. With an appropriate choice of $\epsilon$, the accepted $U$-maximizing decision coincides with the $V$-maximizing decision — the normative choice. Thus, `GenNOP` allows algorithms to handle the measurable dimension, while humans retain authority over the unmeasurable, value-laden dimension. Figure 2 illustrates how humans and algorithms collaborate under the `GenNOP` framework. See Appendix C for a comparison of `GenNOP` with other paradigms of solving human-centered decision-making problems.

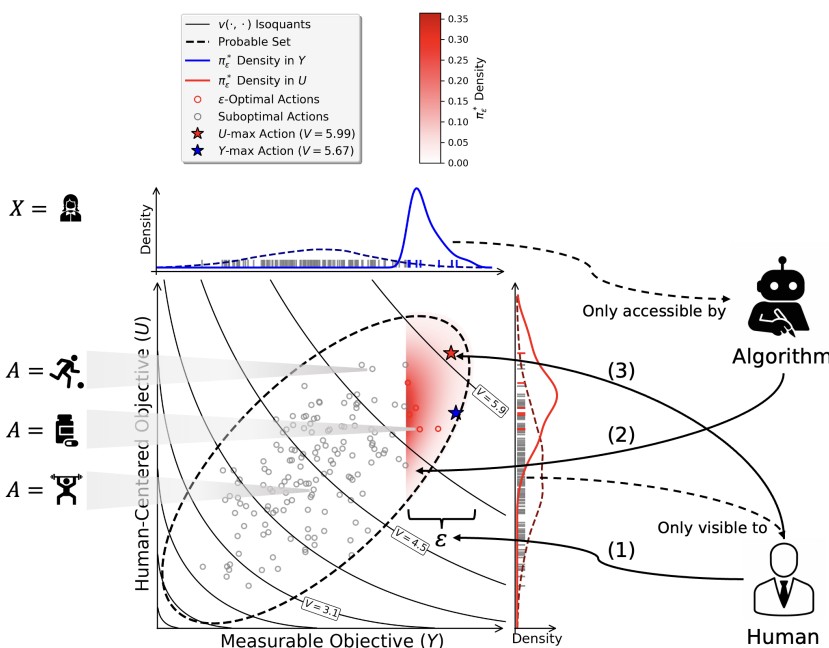

Figure 2: Illustration of Human-Centered Decision-Making Under Generative Near-Optimal Policy Learning (`GenNOP`): (1) Human sets the hyperparameter $\epsilon$; (2) Algorithm learns the distribution of (potentially high-dimensional) actions that are $\epsilon$-optimal, $\pi_\epsilon^*$; (3) Human samples $m$ actions from the learned distribution and chooses the one maximizing the human-centered objective.

**Our Contributions** Our framework offers a natural way of allocating roles to human experts and algorithms, without inducing significant performativity (Perdomo et al., 2020), as human experts are not asked to consider balancing the measurable objective with the human-centered objective—an unnatural task that depends on the algorithmic output—except when they determine the hyperparameter $\epsilon$. We formally define the human-centered decision-making problems in Section 2. In Section 3, we introduce our framework `GenNOP` aimed at solving these problems along with an implementation using max-stable process regression and diffusion models. In Section 4, we showcase

and evaluate our framework and implementation using synthetic and real datasets. See Appendix A for a review of related work.

## 2 HUMAN-CENTERED DECISION-MAKING

**Problem Setup**   We assume access to $n$ i.i.d. observations $\{(\mathbf{a}_i, \mathbf{x}_i, y_i)\}_{i=1}^n \sim \mathcal{D}$ from an offline dataset with covariates $\mathbf{x}_i \in \mathcal{X}$ that characterize individual $i$, action $\mathbf{a}_i \in \mathcal{A}$ taken by individual $i$, and measurable objective value $y_i \in \mathbb{R}$. Note that $\mathbf{x}_i, \mathbf{a}_i$ are vectors, as GenNOP admits multi-dimensional covariates and actions. For notational convenience, we use $x_i, a_i$ thereafter in place of vector notations. We adopt the potential outcomes framework (Rubin, 1974; Imbens & Rubin, 2015).

Formally, our hybrid decision-maker solves a utility-maximization problem with regard to the chosen decision (or *action*) $a$:

$$\max_a V_a, \tag{1}$$

where $V_a = v(Y_a, U_a)$ is the overall utility (or *value*) of an action, $Y_a = y(a)$ the measurable objective, and $U_a = u(a)$ the human-centered objective. The decision-maker does not have full knowledge about the shape of $v(\cdot, \cdot)$ but can make mild assumptions about it.

Unlike the conventional *human-agnostic* setup which aims at optimizing for $Y$, under the *human-centered* setup, the goal (as in Equation (1)) is to optimize for $V$, the overall utility of an action as a function of the measurable objective $Y$ and the human-centered objective $U$, by incorporating both the observed dataset $\mathcal{D}$ and a human evaluator in the loop.

$\epsilon$-**Optimality vs Quantitative Optimality**   If there is some $a^*$ such that $a^* = \arg\max_a Y_a = \arg\max_a U_a$, intuitively, we should never sacrifice quantitative optimality in $Y$. In practice, however, $Y_a$ and $U_a$ often exhibit inherent trade-offs at or near $Y$- and $U$-optimalities: near the $Y$-optimality, small gains in $Y$ often come with significant losses in $U$, and vice versa. For example, if $Y$ represents the commercial success of a movie (measured in box-office revenue) and $U$ its artistic value, modifying the plot of a movie expected to be hugely commercially successful to bring in yet more revenue would require the movie to appeal to an even broader audience at a cost of its artistic value. Two real-world examples are shown in Appendix D. We thus assume that there exists a strictly concave Pareto frontier (Assumption 1). Since in most real-world cases, $V_a$ exhibits diminishing returns to both $Y_a$ and $U_a$, we can assume that $v(\cdot, \cdot)$ is strictly concave in both its inputs (Assumption 3). Along with other mild assumptions stated in Appendix E.1, we have the following:

**Proposition 1** (Maxima non-coincidence). *Let $\{(Y^{(n)}, U^{(n)})\}_{n=1}^N$ be i.i.d. draws satisfying Assumptions 1 — 3. Define:*

$$M_Y \in \arg\max_{1 \le n \le N} Y^{(n)}, \qquad M_V \in \arg\max_{1 \le n \le N} v\left(Y^{(n)}, U^{(n)}\right),$$

*with arbitrary tie-breaking. Then:*

$$\lim_{N \to \infty} \mathbb{P}(M_Y = M_V) = 0.$$

**Proposition 2** (Local $V$-order dominance). *Under Assumptions 1 — 5, there exists $\bar{\epsilon} \in (0, \epsilon]$ such that for any two sampled points with*

$$Y^{(i)}, Y^{(j)} \in [y^\star - \bar{\epsilon}, y^\star],$$

*we have, writing $V^{(k)} := v\left(Y^{(k)}, U^{(k)}\right)$,*

$$U^{(i)} \ge U^{(j)} \implies V^{(i)} \ge V^{(j)}.$$

*Consequently, within any finite sample restricted to $y \in [y^\star - \bar{\epsilon}, y^\star]$, an $\arg\max U$ is also an $\arg\max V$.*

**Proposition 3** (Global $V$-optimality of $U$-maximizers). *Under Assumptions 1 — 4 and 6, there exists $\bar{\epsilon} \in (0, \epsilon]$ such that, over*

$$\mathcal{N}_{\bar{\epsilon}, \eta} := \left\{(y, u) : y \in [y^\star - \bar{\epsilon}, y^\star], \, g(y) - \eta \le u \le g(y)\right\},$$

*every global maximizer of $U$ is also a global maximizer of $V$:*

$$\arg\max_{(y,u)\in\mathcal{N}_{\bar{\epsilon},\eta}} U \ \subseteq \ \arg\max_{(y,u)\in\mathcal{N}_{\bar{\epsilon},\eta}} v(y,u).$$

*Moreover, for i.i.d. samples supported in $\mathcal{N}_{\bar{\epsilon},\eta}$ with density bounded below as in Assumption 6, if $\hat{k}_m \in \arg\max_{1\leq k\leq m} U^{(k)}$ and $V^{(k)} := v\big(Y^{(k)}, U^{(k)}\big)$, then:*

$$\lim_{m\to\infty} \mathbb{P}\left( \left(Y^{(\hat{k}_m)}, U^{(\hat{k}_m)}\right) \in \arg\max_{(y,u)\in\mathcal{N}_{\bar{\epsilon},\eta}} v(y,u) \right) \ = \ 1.$$

See Appendix E.2 for proofs. We formally define $\epsilon$-optimality of actions for an individual characterized by covariates $x$ as follows:

**Definition 1** ($\epsilon$-optimality). *For some $\epsilon \geq 0$, an action $a \in \mathcal{A}$ is considered $\epsilon$-optimal if*

$$|y^*(x) - \mathbb{E}[Y_a \mid X = x]| \leq \epsilon,$$

*where $y^*(x) = \max_{a\in\mathcal{A}} \mathbb{E}[Y_a \mid X = x]$ is the optimal $Y$-value for an individual with covariates $x$.*

Under this definition, $\epsilon$-optimality can vary from individual to individual. For example, treatments that are considered near-optimal for an otherwise healthy individual are intuitively different from those considered near-optimal for an individual with many comorbidities, even when the treatments are intended for the same condition.

If we know the value of $\epsilon$ a priori, we can query the algorithm to find out $y^*$ and generate a large number of actions $a$ such that $Y_a \geq y^* - \epsilon$. We can then ask the human evaluator to choose the one among the generated actions that maximizes $U$: among the generated actions, any $U$-maximizing action is guaranteed to be a $V$-maximizing action. Unfortunately, the human decision-maker does not have explicit access to the value of $\epsilon$, at least not without knowing the shape of $v(\cdot, \cdot)$. However, compared to estimating the shape of $v$, directly estimating $\epsilon$ as a hyperparameter is a much easier and more intuitive task. Even when $\epsilon$ is underestimated, any $U$-maximizing action chosen from the generated actions is still better than the $Y$-maximizing action.

## 3 GENERATIVE NEAR-OPTIMAL POLICY LEARNING (GENNOP)

Our proposed framework, GenNOP, aims at using a conditional generative model to learn a stochastic policy that is $\epsilon$-optimal in $Y$-value. Compared to a deterministic policy that is quantitatively optimal in $Y$-value, an $\epsilon$-optimal stochastic policy offers human evaluators choices from which a higher $V$-value (overall utility) is attained. We precisely define our target policy of interest, $\epsilon$-optimal stochastic policies, as follows:

**Definition 2** ($\epsilon$-optimal policies). *An $\epsilon$-optimal policy $\pi_\epsilon^*$ is a stochastic policy that maps covariates $x$ to the uniform distribution of all $\epsilon$-optimal actions:*

$$\pi_\epsilon^*(a|x) = \frac{1}{|\Omega_\epsilon^*(x)|} \mathbf{1}\left\{a \in \Omega_\epsilon^*(x)\right\}, \tag{2}$$

*where $\Omega_\epsilon^*(x) = \{a : |y^*(x) - \mathbb{E}[Y_a \mid X = x]| \leq \epsilon\}$ and $\mathbf{1}\{\cdot\}$ denotes the indicator function.*

Figure 3 illustrates $\epsilon$-optimality and $\epsilon$-optimal policy. Whereas the density of the absolute-optimal policy $p(\pi^*(x))$ represents a point mass in the action space, that of the $\epsilon$-optimal policy $\Omega_\epsilon^*(x)$ represents a richer, more diverse, and potentially multi-modal density over the action space, at the expense of $\epsilon$ in the $Y$-space, which is desirable by practitioners under various settings.

### 3.1 CONDITIONAL GENERATIVE MODEL AS $\pi_\epsilon^*$

The goal of GenNOP is to train a generative model parametrized by $\theta$ whose generative distribution $\pi_\theta$ approximates $\pi_\epsilon^*$. To this end, we define our learning objective as minimizing the Kullback-Leibler (KL) divergence between the target policy $\pi_\epsilon^*$ and the generative policy $\pi_\theta$:

$$\min_\theta L(\theta) = \mathbb{E}_{x\sim p(X)}[D_{\mathrm{KL}}[\pi_\epsilon^*(\cdot|x) \,||\, \pi_\theta(\cdot|x)]].$$

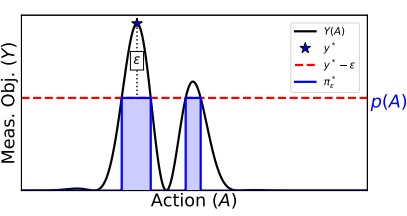



Figure 3: $\epsilon$-Optimal Policy

If we have a dataset $\mathcal{D}^* = \{(x_i, a_i)\}_{i=1}^n$ where $x_i$ is drawn from $p(X)$ and $a_i$ from the true $\epsilon$-optimal policy $\pi_\epsilon^*$, our learning objective above can be conveniently expressed in distribution as minimizing (see Appendix H.1 for derivation):

$$L(\theta) \stackrel{d}{=} \mathbb{E}_{(x,a)\sim\mathcal{D}^*}[-\log \pi_\theta(a|x)]. \tag{3}$$

However, we cannot directly obtain action samples drawn from the target policy $\pi_\epsilon^*$. Instead, we perform an *re-weighting* step to the observational dataset $\mathcal{D}$, so that the distributions of action samples drawn from the re-weighted dataset approximate the distributions of those drawn from $\pi_\epsilon^*$. We rely on two quantities:

1. $g_\epsilon(y, x) = \mathbb{E}_{y^*(x)}[\mathbf{1}\{y^*(x) < y + \epsilon\}] = \mathbb{P}\{y^*(x) < y + \epsilon\}$ is the probability that a given outcome value $y$ is at most $\epsilon$ below the optimal outcome for a individual with covariates $x$. It acts as a *filter* on the observational treatment distribution. Its estimation involves the choice of a parametric distribution and is discussed in Section 3.2.

2. $p(a|x)$ is the generalized propensity score (GPS) of action $a$ given covariates $x$. It is used to transform the filtered *observational* distribution into a *counterfactual* distribution via inverse probability weighting (IPW). Its estimation is discussed in Appendix I.2 by adopting the strategy by Zou et al. (2020).

We define the weight function of an observation given $\epsilon$ as $w(x, a, y; \epsilon) = g_\epsilon(y, x)/p(a|x)$, which is identifiable from observational data under standard causal assumptions in Appendix F. Adopting the conditional diffusion model with classifier-free guidance parametrized by $\theta$ as the generative policy $\pi_\theta$, we re-weight its loss function (6) and have the following learning objective (see Appendix H.2 for derivation):

$$L(\theta) = \mathbb{E}_{t,x,a,y,\varepsilon} \left[ w(x, a, y; \epsilon) \cdot \|\varepsilon - \varepsilon_\theta (a_t, t, x)\|^2 \right]. \tag{4}$$

Overall, our strategy for learning $\pi_\theta$ can be viewed as a two-stage learning process: (1) weight construction via training neural networks that parametrize generalized extreme value (GEV) distributions and estimating the GPS via variational autoencoder (VAE); and (2) conditional diffusion model training with re-weighted objective. In the first stage, we adopt a "filter-and-weight" strategy to construct a re-weighted dataset of uniformly distributed, counterfactually near-optimal actions, out of an observational dataset of self-selected, possibly suboptimal actions. The `GenNOP` algorithm is summarized as Algorithm 1.

## 3.2 ESTIMATING CONDITIONAL OPTIMALITY VIA MAX-STABLE PROCESS REGRESSION

We aim to estimate the probability distribution of the conditional optimality: $g_\epsilon(y, x) = \mathbb{P}\{y^*(x) < y + \epsilon\}$, where $y^*(x) = \max_{a\in\mathcal{A}} \mathbb{E}[Y_a \mid X = x]$. Learning a counterfactual model $f : \mathcal{X} \times \mathcal{A} \to \mathbb{R}$ is intractable for large or continuous $\mathcal{A}$. Instead, note that we need only the value of $\max_{a\in\mathcal{A}} Y_a \mid X = x$, not the $\arg\max$. Hence we view $\{Y_x\}_{x\in\mathcal{X}}$ as a stochastic process, where each $Y_x$ denotes the random variable $\{Y \mid X = x\}$.

Let $\mathcal{X}$ be a metric space with distance $d(\cdot, \cdot)$. For each $x_i$ in our sample, we treat its $kb$ nearest neighbors as having identical covariates $x_i$, randomly partition them into $k$ blocks of size $b$, and take the maximum within each block: $\{y_i^{(1)}, y_i^{(2)}, \ldots, y_i^{(k)}\}$. We repeat this for $i \in \{1, \ldots, n\}$.

---

**Algorithm 1** `Generative Near-Optimal Policy Learning (GenNOP)`

---

**Require:** Dataset $\mathcal{D} = \{(x_i, a_i, y_i)\}_{i=1}^n$, neighbors $k$, block size $b$, metric $d$, threshold $\epsilon$
**Ensure:** $\epsilon$-optimal actions $\{a_1^*, \ldots, a_m^*\}$ for given $x$
 1: Initialize parameters: GEV ($\psi$), VAE ($\phi, \varphi$), diffusion model ($\theta$).
 2: **for** each $x_i \in \mathcal{D}$ **do**
 3:    Find $k \cdot b$ nearest neighbors; partition into $k$ blocks; compute block maxima $\{y_i^{(j)}\}$.
 4: **end for**
 5: Train the GEV model via MLE to obtain $\hat{\psi}$.
 6: **for** each $(x_i, a_i, y_i) \in \mathcal{D}$ **do**
 7:    Compute $g_\epsilon(y_i, x_i) = \mathbb{P}\{y^*(x_i) < y_i + \epsilon\}$ using GEV with $\hat{\psi}$.
 8:    Estimate $p(a_i | x_i)$ via VAE.
 9: **end for**
10: **while** not converged **do**
11:    Sample mini-batch $(x, a, y) \sim \mathcal{D}$, timestep $t$, and noise $\varepsilon$.
12:    Update $\theta$ by minimizing

$$L(\theta) = \mathbb{E}\Big[w(x, a, y; \epsilon) \cdot \|\varepsilon - \varepsilon_\theta(a_t, t, x)\|^2\Big].$$

13: **end while**
14: **Return** $m$ actions sampled from $\pi_\theta(\cdot | x)$.

---

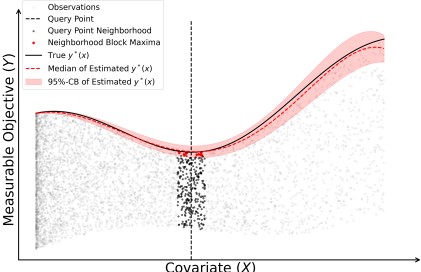

Figure 4: Max-Stable Process Regression

By standard extreme-value theory, the collection $\{y_i^{(j)}\}_{i=1,\ldots,n;\, j=1,\ldots,k}$ admits a max-stable characterization, so each marginal distribution is a GEV distribution with parameters $\mu, \sigma, \xi$, which are estimated via neural networks. See Appendix I.1 for details. Figure 4 illustrates how $y^*(x)$ is estimated probabilistically from block maxima. To showcase the robustness of this method and the effect of different choices of $k, b$, we conducted an ablation study and reported in Table 1 the means and standard deviations of the negative $\log$-likelihood of the estimated parameters over 5 random initializations. We found that overall this method is robust, while moderate regularization strength, number of blocks ($k$), and block size ($b$) can lead to the best performance. See Appendix J for more details.

Table 1: Ablation Study of GEV Parameter Estimation.

| Dimensionality | Regularization | $k = 10, b = 30$ | $k = 20, b = 10$ | $k = 20, b = 20$ | $k = 20, b = 30$ | $k = 30, b = 30$ | $k = 50, b = 30$ |
|---|---|---|---|---|---|---|---|
| 1D | 0 | 0.52 (2.90) | −1.34 (0.16) | −0.56 (1.40) | −0.15 (3.27) | −1.31 (0.66) | −1.24 (0.90) |
| 1D | 1 | −2.06 (0.17) | −1.45 (0.07) | −1.99 (0.13) | −2.17 (0.17) | −2.02 (0.21) | −2.05 (0.07) |
| 1D | 10 | −1.84 (0.31) | −1.19 (0.06) | −1.53 (0.17) | −1.72 (0.55) | −1.49 (0.65) | −1.82 (0.07) |
| 2D | 0 | 3.09 (4.58) | 0.60 (1.41) | 1.07 (2.62) | −0.56 (1.02) | −0.74 (0.88) | 0.29 (3.11) |
| 2D | 1 | −0.95 (0.65) | −0.36 (0.24) | −0.66 (0.44) | −1.22 (0.12) | −1.05 (0.34) | −0.29 (1.56) |
| 2D | 10 | −1.25 (0.13) | −0.33 (0.06) | −0.22 (0.24) | −0.72 (0.42) | −0.68 (0.49) | −0.36 (0.47) |

## 4 EXPERIMENTS

**Synthetic Results** We created the following synthetic datasets to compare `GenNOP` with the baseline methods: (1) A fully-synthetic dataset in which covariates, actions, and measurable objective

values are all 1-dimensional and bounded by $(0, 1)$; (2) A semi-synthetic dataset in which actions are represented by images drawn from the `Fashion-MNIST` dataset to showcase the capability of handling high-dimensional action spaces; covariates in this dataset are also multi-dimensional. Details can be found in Appendix K.

We report the evaluation metrics on the synthetic datasets in the table below. Numbers outside parentheses are the mean metrics taken over the distributions of covariates. Those inside are the 5-th percentile metrics. Standard deviations of the metrics taken over 10 generated samples are indicated by the numbers after $\pm$. 0.00 indicates quantities less than 0.005. Compared to the baseline methods, `GenNOP` enjoys superior performance across metrics and datasets. Moreover, the aim of `GenNOP` to learn policies that give individualized action recommendations is well attained, as the superior performance of `GenNOP` holds not only at the mean but also for its poorest-performing units (indicated by the 5-th percentile covariates). With an acceptable performance even for its poorest-performing units, decision-makers can become more confident in adopting `GenNOP`.

Table 2: Evaluation Metrics.

| | Fully-synthetic dataset | | Semi-synthetic dataset | | |
| Method | Precision ↑ | Recall ↑ | Precision ↑ | Recall ↑ | FID ↓ |
|---|---|---|---|---|---|
| GenNOP | $0.85 \pm 0.00\,(0.47 \pm 0.00)$ | $0.97 \pm 0.00\,(0.80 \pm 0.00)$ | $0.83 \pm 0.03\,(0.38 \pm 0.06)$ | $0.69 \pm 0.03\,(0.27 \pm 0.03)$ | $5.0 \pm 1.8$ |
| GenNOP w/o $p$ | $0.83 \pm 0.00\,(0.37 \pm 0.01)$ | $0.94 \pm 0.00\,(0.80 \pm 0.00)$ | $0.87 \pm 0.03\,(0.44 \pm 0.08)$ | $0.67 \pm 0.02\,(0.25 \pm 0.03)$ | $6.5 \pm 3.4$ |
| GenNOP w/o $g_\epsilon$ | $0.01 \pm 0.00\,(0.02 \pm 0.00)$ | $0.45 \pm 0.00\,(0.10 \pm 0.00)$ | $0.38 \pm 0.02\,(0.00 \pm 0.00)$ | $0.45 \pm 0.02\,(0.00 \pm 0.00)$ | $14.3 \pm 3.8$ |
| DRPolicyForest (Athey & Wager, 2021) | $0.37 \pm 0.00\,(0.00 \pm 0.00)$ | $0.06 \pm 0.00\,(0.00 \pm 0.00)$ | $0.44 \pm 0.02\,(0.00 \pm 0.00)$ | $0.33 \pm 0.03\,(0.00 \pm 0.00)$ | $22.0 \pm 3.3$ |
| DDOM (Krishnamoorthy et al., 2023) | $0.29 \pm 0.00\,(0.00 \pm 0.00)$ | $0.11 \pm 0.00\,(0.00 \pm 0.00)$ | $0.66 \pm 0.04\,(0.04 \pm 0.08)$ | $0.54 \pm 0.03\,(0.04 \pm 0.09)$ | $34.7 \pm 12.4$ |
| GP-UCB (Srinivas et al., 2010) | $0.13 \pm 0.00\,(0.00 \pm 0.00)$ | $0.27 \pm 0.00\,(0.00 \pm 0.00)$ | $0.40 \pm 0.03\,(0.00 \pm 0.00)$ | $0.43 \pm 0.05\,(0.00 \pm 0.00)$ | $19.1 \pm 7.3$ |

The impact of the key quantities $g_\epsilon, p$ is demonstrated by the ablation studies. Without the "filter" $g_\epsilon$, `GenNOP` saw a considerable reduction in performance, particularly in precision, as it learns from all past actions in the training dataset, which can include suboptimal ones. Without the "weight" $p$, `GenNOP` took a relatively minor but still statistically significant performance reduction in most metrics, as the observational distribution of actions can be very different from the uniform distribution.

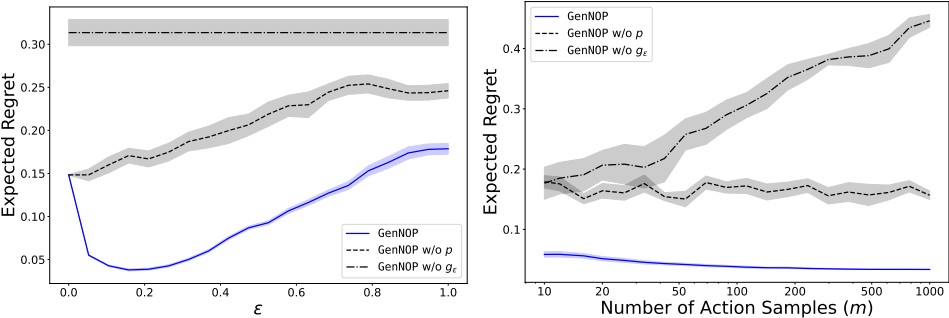

Figure 5: Effect of Hyperparameters on Expected Regret

To bring $\epsilon$-optimal policy into the context of human-centered decision-making, we created an end-to-end synthetic dataset by including $U, V$ in its data-generating process. We define the regret of an action $a$ for an individual with covariates $x$ as the difference between the optimal overall utility value ($V$) for that individual and the overall utility value attained by that action: $\text{Regret}(a, x) = v^*(x) - V_a | X = x$, where $v^*(x) = \max_a V_a | X = x$; additionally, we define the regret of a stochastic policy $\pi(x)$ given the number of draws $m$ as the difference between $v^*(x)$ and the $U$-maximizing action among the $m$ draws:

$$\text{Regret}(\pi, x; m) = v^*(x) - V_{a^*} | X = x,$$

where $a^* = \arg\max_a U_a | X = x, a \in \{a_i\}_{i=1}^m \sim \pi(x)$. We examine how the choices of $\epsilon$ and $m$ influence the expected regret: $\mathbb{E}_x \text{Regret}(\pi_\epsilon, x; m)$ in Figure 5. For `GenNOP`, expected regret decreases as $\epsilon$ increases from 0 up to its optimal value as determined by the shape of the utility function. Past the optimal $\epsilon$ value, expected regret increases; nevertheless, expected regret stays below its value at $\epsilon = 0$ until a very high value of $\epsilon$, indicating the advantage of $\epsilon$-optimality over quantitative optimality ($\epsilon = 0$) as long as a moderate $\epsilon$ is chosen. Given the optimal $\epsilon$, expected regret is already near its minimum even when only a few action samples are generated, indicating

the practicality of `GenNOP`, as human experts can only evaluate a small number of action candidates in practice.

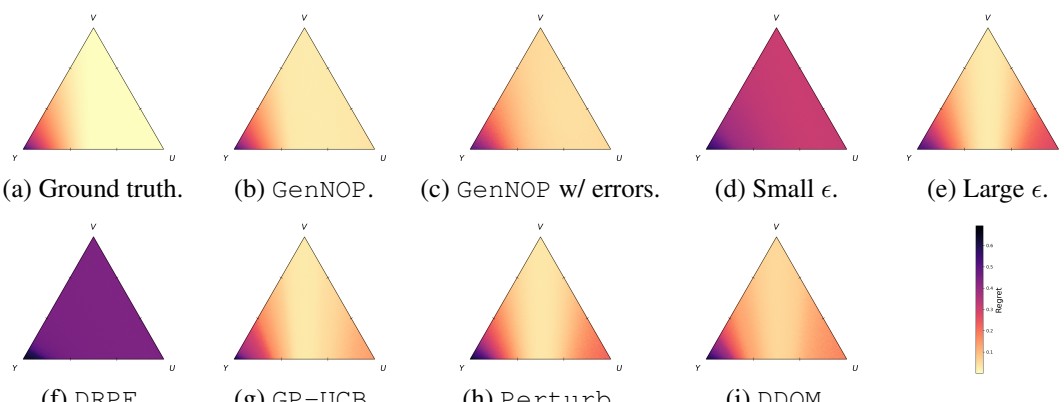

(a) Ground truth.  (b) `GenNOP`.  (c) `GenNOP` w/ errors.  (d) Small $\epsilon$.  (e) Large $\epsilon$.

(f) `DRPF`.  (g) `GP-UCB`.  (h) `Perturb`.  (i) `DDOM`.

Figure 6: Regrets Under Varying Decision-Maker Perception Preferences and Capabilities.

**Decision-Maker Perception Preferences and Capabilities** While we initially assumed decision-makers using `GenNOP` would optimize for the human-centered objective ($U$), in practice different decision-makers have varying focuses: family members prioritize $U$, junior clinicians focus on measurable outcomes ($Y$), and experienced clinicians may optimize for overall utility ($V$). Since both $U$ and $V$ are unmeasurable, we can only qualitatively assess decision-maker preferences, but `GenNOP` should perform well across these diverse practical settings.

To evaluate the decisions made by the human-algorithm hybrid system, we model the human decision-makers as capable of perceiving the quality of any action ($Q_a$) as a linear combination of their perceived $Y_a, U_a, V_a$ values. Given a policy $\pi$, they solve the following optimization problem as their decision:

$$\arg\max_{a \sim \pi} \lambda_Y Y_a (1 + \delta_Y) + \lambda_U U_a (1 + \delta_U) + \lambda_V V_a (1 + \delta_V),$$

where $\lambda_Y + \lambda_U + \lambda_V = 1$ and $\lambda_Y, \lambda_U, \lambda_V \geq 0$; and $\delta_Y, \delta_U, \delta_V \sim \text{Unif}(-\delta, \delta), \delta \geq 0$ are drawn independently to model the limitations in decision-maker perception capabilities.

We assess the quality of the decisions made by the hybrid system by comparing the overall objective of the chosen decision and that of the oracle decision and calculating the regret. Using barycentric coordinates, we plot the regret against the ternary preferences ($\lambda_Y, \lambda_U, \lambda_V$) in equilateral triangles in Figure 6. The ground-truth $\epsilon$-optimal policy achieves zero regret at $(0, 1, 0)$ and trivially at $(0, 0, 1)$, while the algorithm-only system at $(1, 0, 0)$ yields maximum regret under $\delta = 0$. Zero regret remains achievable when $\lambda_Y$ stays below a threshold determined by the $\lambda_U/\lambda_V$ ratio, suggesting decision-makers should prioritize human-centered over overall objectives when facing moderate perception preferences for measurable objectives. While `GenNOP` cannot achieve perfect zero regret due to non-zero densities where $Y_a < y^* - \epsilon$, it maintains low regret except near the $Y$-vertex and significantly outperforms the baseline `DDOM`. Performance depends critically on $\epsilon$ selection: small values ($\epsilon = 0.05$) enable low-to-moderate regret outside the $Y$-vertex neighborhood at higher minimum regret cost, while large values ($\epsilon = 0.5$) resemble `DDOM` with elevated minimum regret, thus recommending conservative $\epsilon$ choices. When perception capabilities are limited by noise ($\delta = 0.2$, with perceived quality multiplied by factors from $[0.8, 1.2]$), `GenNOP` demonstrates reasonable robustness with only slight increases in minimum regret and undesirable region size near the $Y$-vertex. We elaborate further in Appendix L.

**Real Datasets** We apply our framework to the dosing problem mentioned in the Introduction, which is a good example of a human-algorithm hybrid system solving a human-centered decision-making problem. To this end, we extracted 2 datasets from the Medical Information Mart for Intensive Care (MIMIC)-IV (Johnson et al., 2023) dataset: (1) `mimic-icu-cardio` and (2) `mimic-icu-sepsis`, which contain dosages of sets of medications ($A$), patient characteristics ($X$), and measurable objectives ($Y$) of patients admitted to ICUs for cardiovascular and sepsis diagnoses. Details can be found in Appendix K.4.

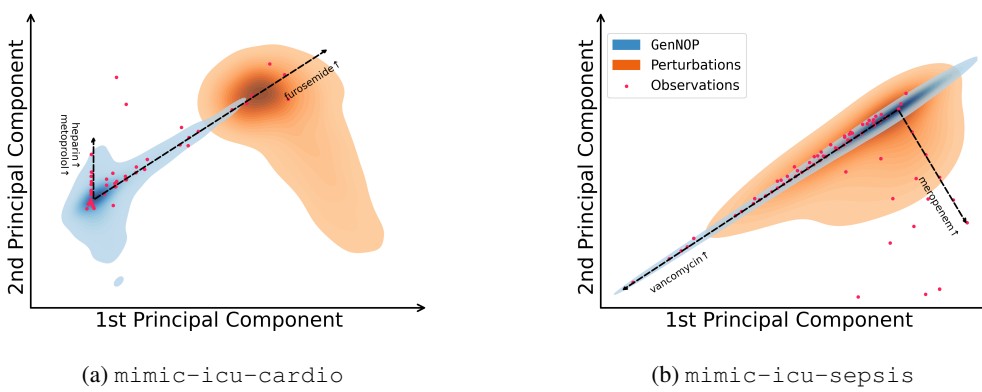

(a) `mimic-icu-cardio`   (b) `mimic-icu-sepsis`

Figure 7: 2-D Representations of Observed Actions and Generated Action Distributions

For each dataset, we plot the observed actions and the generated action distribution by `GenNOP` as 2-D representations in Figure 7 using principal component analysis (PCA). For comparison, we also show the action distribution induced by Gaussian perturbation of the singular optimal action as a result of solving for quantitative optimality instead of $\epsilon$-optimality. This is to mimic the myopic tendency of human behavior when processing the only option available: to perturb it locally. In both datasets, `GenNOP` yield generative distributions more closely aligned with the observed distributions of $\epsilon$-optimal actions taken by human experts than did the perturbation method. Specifically, in `mimic-icu-cardio`, `GenNOP` yield a distribution that covered most of the observed actions, while the perturbation method concentrate its distribution on only a few observed actions and completely miss the mode where the majority of the observed actions reside; in `mimic-icu-sepsis`, although both methods enjoy good coverage of the observed actions, `GenNOP` correctly concentrate on the main axis of actions, while the perturbation method puts too much density on the minor axis, which can lead to lower precision and potentially algorithmic aversion by human decision-makers as a result.

## 5 DISCUSSION

In this paper, we proposed generative near-optimal policy learning (`GenNOP`). We note that as algorithmic capabilities in decision-making steadily improve to the extent of surpassing human capabilities in many aspects, the applications of these capabilities are largely human-agnostic, even when human experts are an integral part of the pipeline, resulting in worse performance of the hybrid system at best and decisions misaligned with human values and preferences with profound impact at worst. Our framework is a compromise between a more integrated human-algorithm system and an easier-to-operationalize mode of design for complementarity.

We note the limitations of our current evaluation strategies. In the synthetic experiments, we assumed human evaluators are capable of choosing the action maximizing $V_a$ (or $Q_a$) regardless of the set of candidate actions presented to them. In practice, the axiom of independence of irrelevant alternatives (IIA) may not hold. In the real dataset experiments, we resorted to clinicians whose actions we observed in the datasets. Instead, an ideal evaluation for these experiments would involve medical experts as raters for the generated actions under different policies in randomized controlled trials. Our framework works the best when ample observations of past decisions are available. When they are sparse, human expertise can help formulate data-driven set optimization problems in place of the generative model. Moreover, a separate algorithm can be trained to mimic the decision-selection process to reduce the human decision-maker's workload, which puts the latter in the role more of a supervisor than a practitioner. Large language models can be a sensible basis for such a algorithm. We leave the exploration of these areas of improvement to future work.

## REPRODUCIBILITY STATEMENT

**Open Access to Code Repository**   Access to our implementation of `GenNOP` is available at `https://anonymous.4open.science/r/GenNOP/`.

**Compute Resources**   All experiments, with the exception of those on the semi-synthetic dataset, should not require significant compute resources beyond the CPU capabilities of a typical recent personal computer. Specifically, they were conducted using a laptop computer with an Apple M3 Pro chip with 18 GB of memory. Each experiment took an insignificant amount of time (on the order of a few minutes).

The experiments on the semi-synthetic dataset were conducted using a virtual machine from Google Cloud Compute Engine with the following configuration: `n1-standard-8 (8 vCPUs, 30 GB Memory), 1 x NVIDIA Tesla P100, 100 GB Storage`. These experiments (including the repetitions needed to establish statistical significance) took about 8 hours.

## USE OF LARGE LANGUAGE MODELS (LLMs)

**Writing**   We used LLMs to assist with restructuring the flow of content and polishing text at the sentence level. All em dashes (—) in this paper are our own writing.

**Retrieval and Discovery**   We used LLM-based Deep Research agents for literature review. All retrieved papers were manually reviewed for relevance. Additional papers were included manually. The Related Work section was manually written.

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

## A   RELATED WORK

There are four bodies of literature closely related to our work: (1) algorithmic decision-making under ambiguity in human factors, (2) optimization with multiple solutions, and (3) generative models for optimization. Our work attempts at tackling the challenges in (1) under the paradigm of (2) with the methodology of (3).

**Algorithmic Decision-Making Under Ambiguity in Human Factors**   Many (Gabriel, 2020; Nick, 2014; Klingefjord et al., 2024; Truong & Koyejo, 2024) have highlighted the challenges in formalizing the multifaceted human values (Schwartz, 1992). Tackling these challenges by surrogate objectives has many pitfalls and carries much risk (Zhuang & Hadfield-Menell, 2020). Another stream of approaches such as reinforcement learning from human feedback (RLHF) (Ouyang et al., 2022) and contrastive preference learning (CPL) (Hejna et al., 2024) assume that human evaluators are capable of expressing their values through preferences. Alur et al. (2024) introduce a framework leveraging "algorithmic indistinguishability" to identify specific instances where human judgment can improve algorithmic predictions. Others adopt multi-objective and uncertainty set approaches (Zhou et al., 2024; Li & Zhu, 2024; Lin et al., 2024). Our framework takes a hybrid approach: following the division of the HITL decision-maker into two personas: a strategic decision-maker ("human designer")—one that determines the goals of the machine—and a practical decision-makers ("human practitioner")—one that oversees the recommended actions by the machine—in Tschiatschek et al. (2024), we put less emphasis on the human designer faithfully specifying their goal and more on the human practitioner correctly judging decisions given by the machine.

**Optimization with Multiple Solutions**   Our work views machines through optimization. The idea of "fighting uncertainty with uncertainty" (Kashyap, 2016) dates back to the Anscombe–Aumann framework (Anscombe et al., 1963), where lotteries address ambiguity in decision-making (Hoxby & Rockoff, 2004; Wouters et al., 2018; Chan, 2013). Stochastic policies can outperform deterministic ones (Delage et al., 2019). Quality-diversity (QD) algorithms (Mouret & Clune, 2015; Cully & Demiris, 2017) relate to our work but have limitations: they need random variations in action spaces, rely on model-based metrics with performance issues (Maragno et al., 2023), and require intensive computation. Other approaches include simulated annealing (Van Laarhoven et al., 1987; Bertsimas & Tsitsiklis, 1993), large-scale neighborhood search (Ahuja et al., 2002; Pisinger & Ropke, 2019), and multi-objective optimization (Deb et al., 2016). These iterative methods face computational costs and stability-convergence tradeoffs (Chen et al., 2018). Our approach leverages generative models, eliminating iterative search requirements. Outside of optimization, conformal predictions have been explored to facilitate human-AI collaboration by generating a set of predictions and deferring some or all prediction efforts to human (Straitouri et al., 2023; De Toni et al., 2024; Madras et al., 2018; Hullman et al., 2025; Ruggieri & Pugnana, 2025).

**Generative Models for Optimization**   Recent generative model advances have been applied to optimization problems. Generative model-based optimization (GMO) methods (Nguyen et al., 2016; Lu et al., 2018; Guo et al., 2022) enable optimization in high-dimensional spaces by using Bayesian optimization and evolutionary algorithms in latent spaces. However, these risk mode collapse due to over-reliance on latent space structure. Our method, while still assuming low-dimensional latent spaces, directly generates action space samples rather than projecting latent-space interpolations. Another approach uses generative models for BBO (Krishnamoorthy et al., 2022; Li et al., 2024), applying conditional generative models with outcomes as conditions.

# B EXAMPLES OF HUMAN-CENTERED DECISION-MAKING PROBLEMS

Table 3: Examples of Human-Centered Decision-Making Problems

| Domain | Key Decision ($A$) | Measurable Objective ($Y$) | Human-Centered Objective ($U$) | Overall Utility ($V$) | Illustrative Trade-Off |
|---|---|---|---|---|---|
| Corporate Hiring | Selecting a new employee. | Candidate's quantifiable metrics (years of experience, test scores, keyword match). | Candidate's unmeasurable qualities (cultural fit, potential, team chemistry). | A productive, collaborative, long-term team member. | Rejecting a high-potential candidate (high $U$) because their resume lacks a specific keyword (low $Y$). |
| Urban Planning | Approving a new development project. | Hard metrics (housing density, tax revenue, traffic flow). | Qualitative experience (neighborhood character, sense of community, aesthetics). | A vibrant, equitable, and livable city. | Maximizing housing density (high $Y$) at the cost of destroying a beloved historic district (low $U$). |
| Product Design | Designing a software feature. | Business KPIs (conversion rate, daily active users, click-through rate). | User's subjective experience (delight, trust, intuitive flow). | A successful product with deep user loyalty. | Using a "dark pattern" to boost sign-ups (high $Y$) while eroding user trust (low $U$). |
| University Admissions | Selecting an incoming class. | Standardized metrics (GPA, SAT/ACT scores, class rank). | Applicant's ineffable potential (leadership, creativity, resilience). | A diverse and dynamic student body that succeeds post-graduation. | Admitting a "test-taker" with perfect scores (high $Y$) over a creative leader with a unique story (high $U$). |
| Financial Investing | Constructing an investment portfolio. | Financial performance indicators (ROI, alpha, Sharpe ratio). | Ethical and moral alignment (ESG principles, social impact, personal values). | A portfolio that generates wealth and provides peace of mind. | Forgoing a highly profitable but unethical investment (high $Y$, low $U$). |
| Film Curation | Acquiring a film for distribution. | Commercial performance (gross box office revenue). | Artistic merit and critical acclaim (story quality, creative vision). | A film that is both a commercial and cultural success. | Green-lighting a formulaic sequel (high $Y$ potential) over an innovative indie film (high $U$ potential). |
| Personal Well-being | Choosing a meal on a diet. | Nutritional data (low sugar content, low calories). | Subjective experience (perceived sweetness, palatability, satisfaction). | A healthy and satisfying food choice that supports long-term adherence. | Eating a nutritionally perfect but tasteless meal (high $Y$, low $U$), leading to later cravings and diet failure. |

## C    DIFFERENT PARADIGMS OF SOLVING HUMAN-CENTERED DECISION-MAKING PROBLEMS

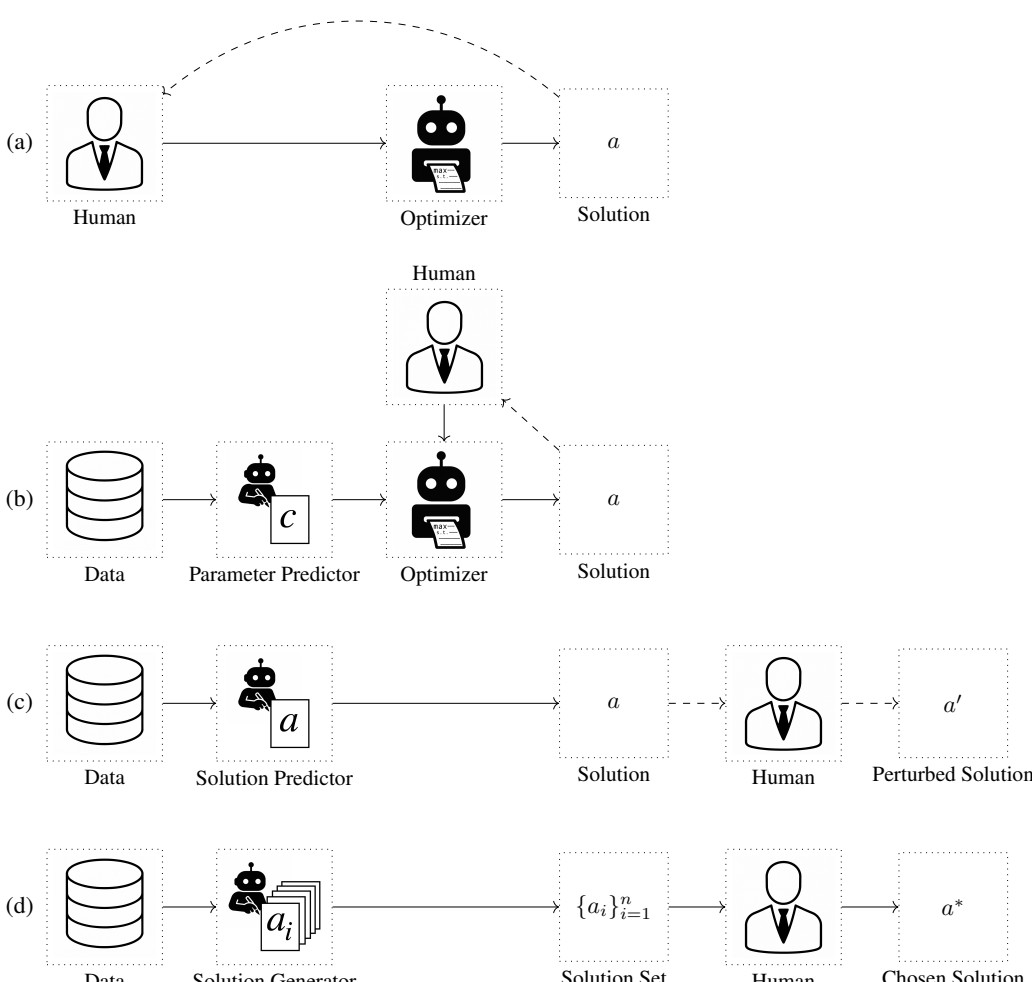

Figure 8: Illustration of Different Paradigms of Solving Human-Centered Decision-Making Problems

(a) Human-programmed optimization; (b) Data-driven optimization with human input; (c) Direct learning; (d) Generative near-optimal policy learning (`GenNOP`). Solid arrows indicate primary procedures; dashed arrows indicate secondary procedures if the solution from a primary procedure is not accepted by a human evaluator: in (a) and (b), human programmers have to readjust the optimization parameters, a hard task involving human prediction of optimizer behavior; in (c), human evaluators make localized perturbations to the solution as they have no control over predictor behavior. `GenNOP` is the only paradigm that does not involve any secondary procedure as it explicitly allows human evaluators to express their tacit knowledge by presenting them multiple solutions.

# D    REAL-WORLD EXAMPLES OF THE $Y_a - U_a$ RELATIONSHIP

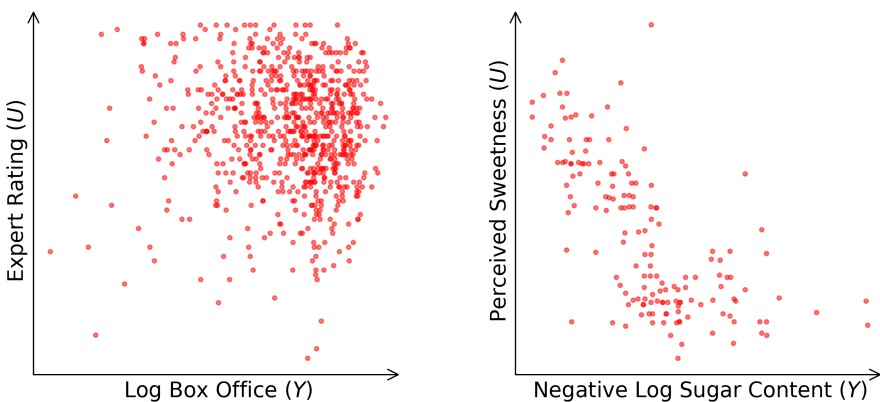

Figure 9: Real-World Examples of the $Y_a - U_a$ Relationship

(a) The left figure illustrates a scenario where an expert movie curator recommends a movie title to a general consumer: the curator believes both the commercial success and the artistic value of a movie positively contribute to the general consumer's utility. The measurable objective ($Y_a$) is represented by the $\log$ of the gross box office; the human-centered objective ($U_a$) is represented by the IMDb Metascore, a proxy for what the curator in our scenario would evaluate the artistic value of the movies. Here $Y_a$ and $U_a$ have a weak, positive correlation. The movie with the highest box office is not the one with the highest expert rating. (b) The right figure illustrates a scenario where a sweet-tooth consumer on a low-calorie diet makes a food choice: the consumer's utility depends on both the lack of sugar content (the measurable objective, $Y_a$) and the perceived sweetness (the human-centered objective, $U_a$) of the foods. Here $Y_a$ and $U_a$ have a negative correlation.

# E    ASSUMPTIONS AND PROOFS OF SECTION 2

## E.1    ASSUMPTIONS

**Assumption 1** (Bounded support with strictly concave frontier)**.** *There exist $y_- < y_+$ and a continuous, strictly concave $g : [y_-, y_+] \to \mathbb{R}$ such that*

$$\mathbb{P}\big((Y, U) \in S\big) = 1, \qquad S := \{(y, u) : y \in [y_-, y_+],\ u \leq g(y)\},$$

*and the Pareto frontier is $\{(y, g(y)) : y \in [y_-, y_+]\}$.*

**Assumption 2** (Thick near-frontier band)**.** *There exist $\eta > 0$ and $c > 0$ such that $(Y, U)$ admits a density $f$ with*

$$f(y, u) \geq c \quad \text{for all } y \in [y_-, y_+] \text{ and } g(y) - \eta \leq u \leq g(y).$$

**Assumption 3** (Monotone aggregator with interior $V$-maximizer)**.** *The aggregator $v(\cdot, \cdot) : \mathbb{R}^2 \to \mathbb{R}$ is continuous and strictly increasing in each argument, and with $V(y, u) = v(y, u)$ there is a unique maximizer*

$$(y^\star, u^\star) \in \arg\max_{(y,u) \in S} V(y, u) \quad \text{with } u^\star = g(y^\star),\ y^\star \in (y_-, y_+).$$

**Assumption 4** (Local tradeoff bounds for $v(\cdot, \cdot)$ near $(y^\star, g(y^\star))$)**.** *There exist $\epsilon_0 > 0$, $\eta_0 > 0$ and constants $L_Y, m_U > 0$ such that for all*

$$y, y' \in [y^\star - \epsilon_0, y^\star], \quad u, u' \in [g(y) - \eta_0, g(y)],$$

$$v(y', u) - v(y, u) \leq L_Y |y' - y|, \qquad v(y, u') - v(y, u) \geq m_U (u' - u).$$

**Assumption 5** (Local sampling restriction). *For some $\epsilon \in (0, \epsilon_0]$ and $\eta \in (0, \eta_0]$, we draw i.i.d. candidates $(Y^{(k)}, U^{(k)})$ satisfying*

$$\mathbb{P}\Big((Y^{(k)}, U^{(k)}) \in \mathcal{N}_{\epsilon,\eta}\Big) = 1, \qquad \mathcal{N}_{\epsilon,\eta} := \{(y, u) : y \in [y^\star - \epsilon, y^\star], \ g(y) - \eta \leq u \leq g(y)\}.$$

**Assumption 6** (Global coverage on the frontier neighborhood). *For the same $\epsilon, \eta$ as above, the sampling distribution has a density $f_{\text{samp}}$ with*

$$f_{\text{samp}}(y, u) \ \geq \ c_{\text{samp}} > 0 \ \text{for all} \ (y, u) \in \mathcal{N}_{\epsilon,\eta}, \qquad f_{\text{samp}}(y, u) = 0 \ \text{outside} \ \mathcal{N}_{\epsilon,\eta}.$$

### E.2 PROOFS

*Proof of Proposition 1.* By Assumption 2, the marginal of $Y$ has positive density on a right–endpoint neighborhood of $y_+$. Hence, for any $\varepsilon > 0$,

$$\mathbb{P}\left(Y \in [y_+ - \varepsilon, \, y_+]\right) \ \geq \ c\,\varepsilon > 0,$$

and the sample maximum satisfies $Y^{(M_Y)} \to y_+$ almost surely as $N \to \infty$.

By Assumptions 1 and 3, the unique population maximizer of $V(y, u) = v(y, u)$ over the compact set $S$ is $(y^\star, g(y^\star))$ with $y^\star \in (y_-, y_+)$. Uniqueness and continuity yield: for every $\rho > 0$ there exists $\kappa(\rho) > 0$ such that

$$\sup_{\substack{(y,u) \in S \\ \|(y,u)-(y^\star, g(y^\star))\| \geq \rho}} V(y, u) \ \leq \ V(y^\star, g(y^\star)) - \kappa(\rho).$$

Since, by Assumption 2, any neighborhood of $(y^\star, g(y^\star))$ inside $S$ has positive probability, the empirical maximizer satisfies

$$\Big(Y^{(M_V)}, U^{(M_V)}\Big) \ \xrightarrow{p} \ (y^\star, g(y^\star)).$$

Fix $\varepsilon \in \left(0, \frac{y_+ - y^\star}{3}\right)$. With probability tending to one,

$$Y^{(M_Y)} \geq y_+ - \varepsilon \qquad \text{and} \qquad Y^{(M_V)} \leq y^\star + \varepsilon < y_+ - 2\varepsilon,$$

so $Y^{(M_V)} < Y^{(M_Y)}$ and hence $M_V \neq M_Y$. Therefore $\mathbb{P}(M_Y = M_V) \to 0$. $\qquad \square$

*Proof of Proposition 2.* Work on the event (which has probability one under absolutely continuous sampling) that there are no exact ties in the $U$-coordinates among the finitely many sampled points. Let $i, j$ be two sampled indices with

$$Y^{(i)}, Y^{(j)} \in [y^\star - \epsilon, \, y^\star] \quad \text{and} \quad U^{(i)} \geq U^{(j)}.$$

By Assumption 4, for any such pair we can write

$$V^{(i)} - V^{(j)} = \left[v\Big(Y^{(i)}, U^{(i)}\Big) - v\Big(Y^{(j)}, U^{(i)}\Big)\right] + \left[v\Big(Y^{(j)}, U^{(i)}\Big) - v\Big(Y^{(j)}, U^{(j)}\Big)\right]$$

$$\geq -L_Y \left|Y^{(i)} - Y^{(j)}\right| + m_U \left(U^{(i)} - U^{(j)}\right).$$

Let $\Delta_U^{\min} := \min\left\{U^{(i)} - U^{(j)} : U^{(i)} > U^{(j)}\right\}$ over the (finite) sample; on the no-tie event, $\Delta_U^{\min} > 0$. Choose

$$\bar{\epsilon} := \min\left\{\epsilon, \ \frac{m_U}{L_Y} \Delta_U^{\min}\right\}.$$

Then for any $i, j$ with $Y^{(i)}, Y^{(j)} \in [y^\star - \bar{\epsilon}, \, y^\star]$ and $U^{(i)} > U^{(j)}$ we have

$$V^{(i)} - V^{(j)} \ \geq \ -L_Y \bar{\epsilon} + m_U \Delta_U^{\min} \ \geq \ 0.$$

If $U^{(i)} = U^{(j)}$ (a null event under absolute continuity), the conclusion $V^{(i)} \geq V^{(j)}$ holds whenever $Y^{(i)} \geq Y^{(j)}$ by monotonicity of $v(\cdot, \cdot)$ in its first argument. Hence, almost surely, the stated implication holds for all pairs, and in particular any $\arg\max U$ over $\left\{k : Y^{(k)} \in [y^\star - \bar{\epsilon}, \, y^\star]\right\}$ is also an $\arg\max V$. $\qquad \square$

*Proof of Proposition 3.* By Assumptions 1 and 3, $(y^\star, g(y^\star))$ is the unique maximizer of $V$ over $S$. Hence, by continuity and strict optimality, there exists $\epsilon_1 \in (0, \epsilon]$ and $\kappa > 0$ such that

$$V(y^\star, g(y^\star)) \geq V(y, g(y)) + 2\kappa \qquad \forall\, y \in [y^\star - \epsilon_1,\, y^\star). \tag{5}$$

By Assumption 4, for all $y \in [y^\star - \epsilon_1,\, y^\star]$ and all $u \leq g(y)$,

$$V(y, g(y)) - V(y, u) \geq m_U\, (g(y) - u) \geq 0.$$

Combining with (5) gives, for all $(y, u) \in \mathcal{N}_{\epsilon_1, \eta}$ with $y < y^\star$,

$$V(y^\star, g(y^\star)) - V(y, u) \geq 2\kappa - m_U\, (g(y) - u) \geq 2\kappa - m_U\, \eta.$$

Choose $\bar{\epsilon} \in (0, \epsilon_1]$ so that $2\kappa - m_U\, \eta > 0$. Then

$$V(y^\star, g(y^\star)) > \sup\left\{ V(y, u) : (y, u) \in \mathcal{N}_{\bar{\epsilon}, \eta},\ y < y^\star \right\}.$$

Therefore $(y^\star, g(y^\star))$ is the unique $V$-maximizer on $\mathcal{N}_{\bar{\epsilon}, \eta}$. Any $U$-maximizer over $\mathcal{N}_{\bar{\epsilon}, \eta}$ must occur on the frontier, i.e., at some $(y, g(y))$ with $y \in [y^\star - \bar{\epsilon},\, y^\star]$. If $y < y^\star$, the above inequality shows it cannot maximize $V$; consequently any $U$-maximizer must be at $y = y^\star$, hence is also a $V$-maximizer. This proves

$$\arg\max_{(y,u) \in \mathcal{N}_{\bar{\epsilon}, \eta}} U \ \subseteq\ \arg\max_{(y,u) \in \mathcal{N}_{\bar{\epsilon}, \eta}} V(y, u).$$

For the sampling statement, under Assumption 6 the i.i.d. sample has density bounded below on $\mathcal{N}_{\bar{\epsilon}, \eta}$, so the empirical $\arg\max U$ converges almost surely to the set $\arg\max_{\mathcal{N}_{\bar{\epsilon}, \eta}} U$, which we have just shown is $\{(y^\star, g(y^\star))\}$. Hence, if $\hat{k}_m \in \arg\max_{1 \leq k \leq m} U^{(k)}$,

$$\mathbb{P}\left( \left(Y^{(\hat{k}_m)}, U^{(\hat{k}_m)}\right) \in \arg\max_{(y,u) \in \mathcal{N}_{\bar{\epsilon}, \eta}} V(y, u) \right) \longrightarrow 1. \qquad \square$$

# F  ASSUMPTIONS IN SECTION 3

We adopt the potential outcomes framework (Rubin, 1974; Imbens & Rubin, 2015). We write $Y_a$ as the potential outcome for the measurable objective of an individual taken action $a$ and make the following standard assumptions:

1. *Consistency*: Provided the action is $a$, then $Y_a$ is the potential outcome under action **a**. Formally, $A = a$ implies $Y_a = Y$.

2. *Unconfoundedness*: $Y_a \perp\!\!\!\perp A \mid X$ for all $a$. This assumption implies action selection is as good as randomized given covariates.

3. *Positivity*: Every individual has a non-zero chance of taking any action in $\mathcal{A}$, namely, $p(a|x) > 0$ for all $a$. $p(a|x)$ denotes the GPS (Imbens, 2000; Hirano & Imbens, 2004).

# G  PRELIMINARIES

**Diffusion Models**  Originally introduced by Ho et al. (2020), diffusion models have proven effective in generating high-dimensional data such as images and videos (Rombach et al., 2022; Ho et al., 2022). Diffusion models progressively corrupt data with noise and then learn to reverse this process. In our context, a neural network parametrized by $\theta$ is trained to predict the noise added to an action, similar to how these models generate images.

To generate a new action, we begin with a sample drawn from a standard Gaussian distribution. Then, using the learned reverse process, we iteratively update the noisy action. At each reverse step $t$, the network computes a noise prediction $\tilde{\varepsilon}_\theta(a_t, t, x)$ and updates the action as $a_{t-1} = \frac{1}{\sqrt{1-\gamma_t}}\left[a_t - \gamma_t\, \tilde{\varepsilon}_\theta(a_t, t, x)\right] + \sqrt{\gamma_t}\, z_t$, where $z_t \sim \mathcal{N}(0, I)$. After reversing all steps down to $t = 0$, the resulting $a_0$ is the generated policy sample, which can be conditioned on $x$ if desired.

Training is performed by randomly selecting a time step $t$ and corrupting the original action $a_0$ with Gaussian noise $a_t = \sqrt{\bar{\lambda}_t}\, a_0 + \sqrt{1 - \bar{\lambda}_t}\, \varepsilon,\quad \varepsilon \sim \mathcal{N}(0, I)$, where $\bar{\lambda}_t = \prod_{s=1}^{t}(1-\gamma_s)$. The network parameters $\theta$ are optimized by minimizing the mean squared error between the actual noise $\varepsilon$ and the predicted noise $\varepsilon_\theta(a_t, t, x)$:

$$L(\theta) = \mathbb{E}_{t, a_0, \varepsilon}\, \|\varepsilon - \varepsilon_\theta(a_t, t, x)\|^2. \tag{6}$$

**Max-Stable Processes**  Let $Y_x^{(1)}, \ldots, Y_x^{(k)}$ be a sequence of $k$ independent copies of a stochastic process $\{Y_x : x \in \mathcal{X}\}$. Define the rescaled pointwise maximum with functions $c_k$ and $d_k$ as:

$$Y_x^* = \left[ \max_{i=1,\ldots,k} Y_x^{(i)} - d_k(x) \right] \Big/ c_k(x), \quad x \in \mathcal{X},$$

If there are sequences of functions $c_k(x) > 0$ and $d_k(x) \in \mathbb{R}$ such that for all $k \in \mathbb{N}$, $\{Y_x^*\}_{x \in \mathcal{X}} \stackrel{d}{=} \{Y_x\}_{x \in \mathcal{X}}$, then $\{Y_x\}_{x \in \mathcal{X}}$ is a *max-stable* process. Its marginal distributions are generalized extreme-value (GEV) distributions, which can be expressed parametrically. The log-likelihood of observing $\{y_j\}$ under the GEV distribution parametrized by $\mu, \sigma, \xi$ is given by:

$$\ell\left(\{y^{(j)}\}; \mu, \sigma, \xi\right) = -k \log \sigma(l) - (1 + 1/\xi) \sum_{j=1}^{k} \log \left[ 1 + \xi \left( \frac{y^{(j)} - \mu}{\sigma} \right) \right]$$

$$- \sum_{j=1}^{k} \left[ 1 + \xi \left( \frac{y^{(j)} - \mu}{\sigma} \right) \right]^{-1/\xi},$$

provided that $1 + \xi \left( \frac{y^{(j)} - \mu}{\sigma} \right) > 0$, for $j = 1, \ldots, k$.

# H  GENNOP LEARNING OBJECTIVE DERIVATION

## H.1  DATA-BASED LEARNING OBJECTIVE

To derive Equation (3) from Equation (3.1), we expand the KL divergence:

$$\begin{aligned}
\min_\theta L(\theta) &= \mathbb{E}_{x \sim p(X)}[D_{\mathrm{KL}}[\pi_\epsilon^*(\cdot|x) \,\|\, \pi_\theta(\cdot|x)]] \\
&= \mathbb{E}_{x \sim p(X)} \left[ \mathbb{E}_{a \sim \pi_\epsilon^*(\cdot|x)} \left[ \log \frac{\pi_\epsilon^*(a|x)}{\pi_\theta(a|x)} \right] \right] \\
&= \mathbb{E}_{x \sim p(X)} \left[ \mathbb{E}_{a \sim \pi_\epsilon^*(\cdot|x)} [\log \pi_\epsilon^*(a|x) - \log \pi_\theta(a|x)] \right] \\
&= \mathbb{E}_{x \sim p(X), a \sim \pi_\epsilon^*(\cdot|x)} [\log \pi_\epsilon^*(a|x) - \log \pi_\theta(a|x)] \\
&\stackrel{d}{=} \mathbb{E}_{(x,a) \sim \mathcal{D}^*} [\log \pi_\epsilon^*(a|x) - \log \pi_\theta(a|x)]
\end{aligned}$$

Since $\log \pi_\epsilon^*(a|x)$ does not depend on $\theta$, the above can be equivalently expressed as:

$$\min_\theta \mathbb{E}_{(x,a) \sim \mathcal{D}^*} [-\log \pi_\theta(a|x)].$$

## H.2  RE-WEIGHTED DIFFUSION MODEL LEARNING OBJECTIVE

To derive Equation (4) from Equations (3) and (6), we use importance sampling:

$$L(\theta) = \mathbb{E}_{(x,a)\sim\mathcal{D}^*}[-\log\pi_\theta(a|x)]$$

$$= \mathbb{E}_{(x,a,y)\sim\mathcal{D}}\left[\frac{p_{\mathcal{D}^*}(x,a)}{p_{\mathcal{D}}(x,a,y)}\cdot(-\log\pi_\theta(a|x))\right]$$

$$= \mathbb{E}_{(x,a,y)\sim\mathcal{D}}\left[\frac{\pi_\epsilon^*(a|x)}{p(a|x)}\cdot(-\log\pi_\theta(a|x))\right]$$

$$= \mathbb{E}_{(x,a,y)\sim\mathcal{D}}\left[\frac{g_\epsilon(y,x)}{p(a|x)}\cdot(-\log\pi_\theta(a|x))\right]$$

$$= \mathbb{E}_{(x,a,y)\sim\mathcal{D}}\left[w(x,a,y;\epsilon)\cdot(-\log\pi_\theta(a|x))\right]$$

$$= \mathbb{E}_{(x,a,y)\sim\mathcal{D}}\left[w(x,a,y;\epsilon)\cdot\mathbb{E}_{t,\varepsilon}\left\|\varepsilon-\varepsilon_\theta(a_t,t,x)\right\|^2\right]$$

$$= \mathbb{E}_{(x,a,y)\sim\mathcal{D},t,\varepsilon}\left[w(x,a,y;\epsilon)\cdot\left\|\varepsilon-\varepsilon_\theta(a_t,t,x)\right\|^2\right]$$

$$= \mathbb{E}_{t,x,a,y,\varepsilon}\left[w(x,a,y;\epsilon)\cdot\left\|\varepsilon-\varepsilon_\theta(a_t,t,x)\right\|^2\right]$$

# I  DETAILS IN SECTION 3

## I.1  MAX-STABLE PROCESS REGRESSION

To allow $\mu$, $\sigma$, and $\xi$ to vary with $x$, we write: $\mu(x;\psi_1)$,$\sigma(x;\psi_2)$, $\xi(x;\psi_3)$, where $\psi = \{\psi_1,\psi_2,\psi_3\}$ are parameters of neural networks. The GEV log-likelihood for block maxima at location $x_i$ is:

$$\ell\Big(\{y_i^{(j)}\};\mu(x_i;\psi_1),\sigma(x_i;\psi_2),\xi(x_i;\psi_3)\Big),$$

subject to the positivity constraint $1+\xi(x_i;\psi_3)\frac{y_i^{(j)}-\mu(x_i;\psi_1)}{\sigma(x_i;\psi_2)}>0$ for all $j=1,\ldots,k$. Maximizing the sum of the log-likelihoods over $i=1,\ldots,n$ with respect to $\psi$ yields the MLE parameters $\hat\psi$. With max-stable process regression, we have:

$$g_\epsilon(y,x) = \mathbb{P}\{\text{GEV}\big(\mu(x;\hat\psi_1),\sigma(x;\hat\psi_2),\xi(x;\hat\psi_3)\big)<y+\epsilon\}.$$

Compared to methods that yield point estimates for $y^*(x)$, max-stable process regression offers a probabilistic alternative. With point estimates, the key quantity $g_\epsilon(y,x)$ can only take values in $\{0,1\}$; with probabilistic estimates, it can take values in $[0,1]$, thereby avoiding hard cutoffs. In regions of $X$ where observations are sparse, allowing gradual decay in contribution to the `GenNOP` objective is particularly desirable over hard cutoffs, which make `GenNOP` more sensitive to uncertainties in $y^*(x)$ estimates.

## I.2  ESTIMATING GENERALIZED PROPENSITY SCORES VIA VARIATIONAL SAMPLE WEIGHT LEARNING

Because the policies of our consideration are potentially high-dimensional, traditional approaches to estimating GPS in the denominator will fail. To circumvent this issue, we assume that the policies have a latent low-dimensional representation, denoted as $Z$, and adopt the strategy by Zou et al. (2020) that learns $Z$ via variational autoencoder (VAE) (Kingma, 2013). With encoder and decoder networks parametrized by $\phi$ and $\varphi$, respectively, we maximize the evidence lower bound (ELBO): $L_{\text{ELBO}} = \frac{1}{n}\sum_{i=1}^n\mathbb{E}_{z\sim q_\phi(z|a_i)}[\ell(a_i,z;\varphi,\phi)]$, where $\ell(a_i,z;\varphi,\phi) = \log p_\varphi(a_i|z)+\log p(z)-\log q_\phi(z|a_i)$.

The stabilized weight $p(a_i)/p(a_i|x_i)$ can be expressed as:

$$\frac{p(a_i)}{p(a_i|x_i)} = \frac{1}{\int_z p(z|x_i)\frac{p(a_i|z)}{p(a_i)}\,dz}$$

$$= \frac{1}{\int_z p(z|x_i)\frac{p(z|x_i)}{p(z)}\,dz}$$

$$= \frac{1}{\mathbb{E}_{z \sim q_\phi(z|a_i)}\left[\frac{p(z|x_i)}{p(z)}\right]}.$$

## J    ABLATION STUDY OF GEV PARAMETER ESTIMATION DETAILS

Table 4: Ablation Study of GEV Parameter Estimation (Full).

| Dimensionality | Regularization | $k=10, b=10$ | $k=10, b=20$ | $k=10, b=30$ | $k=20, b=10$ | $k=20, b=20$ | $k=20, b=30$ | $k=30, b=10$ | $k=30, b=20$ | $k=30, b=30$ | $k=50, b=10$ | $k=50, b=20$ | $k=50, b=30$ |
|---|---|---|---|---|---|---|---|---|---|---|---|---|---|
| 1D | 0 | 1.10 (4.63) | 13.98 (16.49) | 0.52 (2.90) | −1.34 (0.16) | −0.56 (1.40) | −0.15 (3.27) | 0.35 (2.81) | −1.38 (0.97) | −1.31 (0.66) | −1.17 (0.48) | −1.81 (0.17) | −1.24 (0.90) |
| 1D | 0.05 | −0.84 (0.58) | 1.58 (4.18) | −0.22 (1.38) | −1.48 (0.28) | 3.30 (10.05) | −1.46 (0.62) | −1.48 (0.15) | −1.31 (0.30) | −1.11 (0.97) | −1.24 (0.29) | −1.54 (0.49) | −2.01 (0.17) |
| 1D | 0.1 | −0.11 (2.15) | −1.62 (0.42) | 2.35 (7.56) | −0.56 (1.51) | −1.45 (0.65) | −0.39 (3.04) | −1.39 (0.19) | −1.39 (0.56) | −0.51 (1.52) | −0.38 (1.88) | −1.90 (0.07) | −1.91 (0.35) |
| 1D | 0.5 | −0.64 (1.17) | 0.81 (5.39) | 0.61 (4.43) | −1.59 (0.07) | −0.95 (1.88) | −2.10 (0.28) | −1.61 (0.06) | −2.05 (0.07) | −1.81 (0.49) | −1.47 (0.11) | −1.33 (0.77) | −2.08 (0.07) |
| 1D | 1 | −1.48 (0.06) | −1.88 (0.22) | −2.06 (0.17) | −1.45 (0.07) | −1.99 (0.13) | −2.17 (0.17) | −1.40 (0.13) | −1.95 (0.09) | −2.02 (0.21) | −1.38 (0.07) | −1.77 (0.19) | −2.05 (0.07) |
| 1D | 10 | −1.13 (0.13) | −1.33 (0.57) | −1.84 (0.31) | −1.19 (0.06) | −1.53 (0.17) | −1.72 (0.55) | −0.85 (0.23) | −1.63 (0.26) | −1.49 (0.65) | −0.66 (0.35) | −1.26 (0.40) | −1.82 (0.07) |
| 2D | 0 | 0.36 (1.61) | −0.61 (0.53) | 3.09 (4.58) | 0.60 (1.41) | 1.07 (2.62) | −0.56 (1.02) | −0.52 (0.48) | −0.94 (0.57) | −0.74 (0.88) | −0.37 (0.61) | −0.59 (0.33) | 0.29 (3.11) |
| 2D | 0.05 | 0.16 (1.24) | −0.92 (0.18) | 1.56 (5.28) | 1.47 (2.62) | −0.85 (0.45) | −0.66 (0.68) | −0.15 (1.13) | −0.87 (0.24) | −1.19 (0.18) | −0.56 (0.39) | −0.75 (0.39) | −1.13 (0.25) |
| 2D | 0.1 | 0.79 (1.34) | 0.21 (1.24) | 3.85 (6.90) | −0.69 (0.16) | −0.71 (0.51) | 0.53 (2.94) | −0.25 (0.40) | −0.95 (0.32) | −0.94 (0.50) | 0.54 (1.23) | −0.86 (0.34) | −0.77 (0.84) |
| 2D | 0.5 | −0.53 (0.50) | −0.44 (1.47) | −1.23 (0.24) | −0.29 (0.95) | −0.26 (1.26) | −1.22 (0.23) | −0.48 (0.38) | −0.92 (0.17) | −1.19 (0.09) | 1.26 (3.35) | −0.81 (0.24) | −1.01 (0.22) |
| 2D | 1 | −0.64 (0.12) | −1.00 (0.26) | −0.95 (0.65) | −0.36 (0.24) | −0.66 (0.44) | −1.22 (0.12) | −0.15 (0.26) | −1.08 (0.04) | −1.05 (0.34) | 10.13 (16.32) | 3.06 (6.88) | −0.29 (1.56) |
| 2D | 10 | −0.16 (0.34) | −0.93 (0.09) | −1.25 (0.13) | −0.33 (0.06) | −0.22 (0.24) | −0.72 (0.42) | 8.66 (17.81) | −0.64 (0.32) | −0.68 (0.49) | 12.41 (18.23) | 4.04 (6.56) | −0.36 (0.47) |

GEV Regression Sensitivity: Num Blocks ($k$) vs Block Size ($b$) [Reg=0]

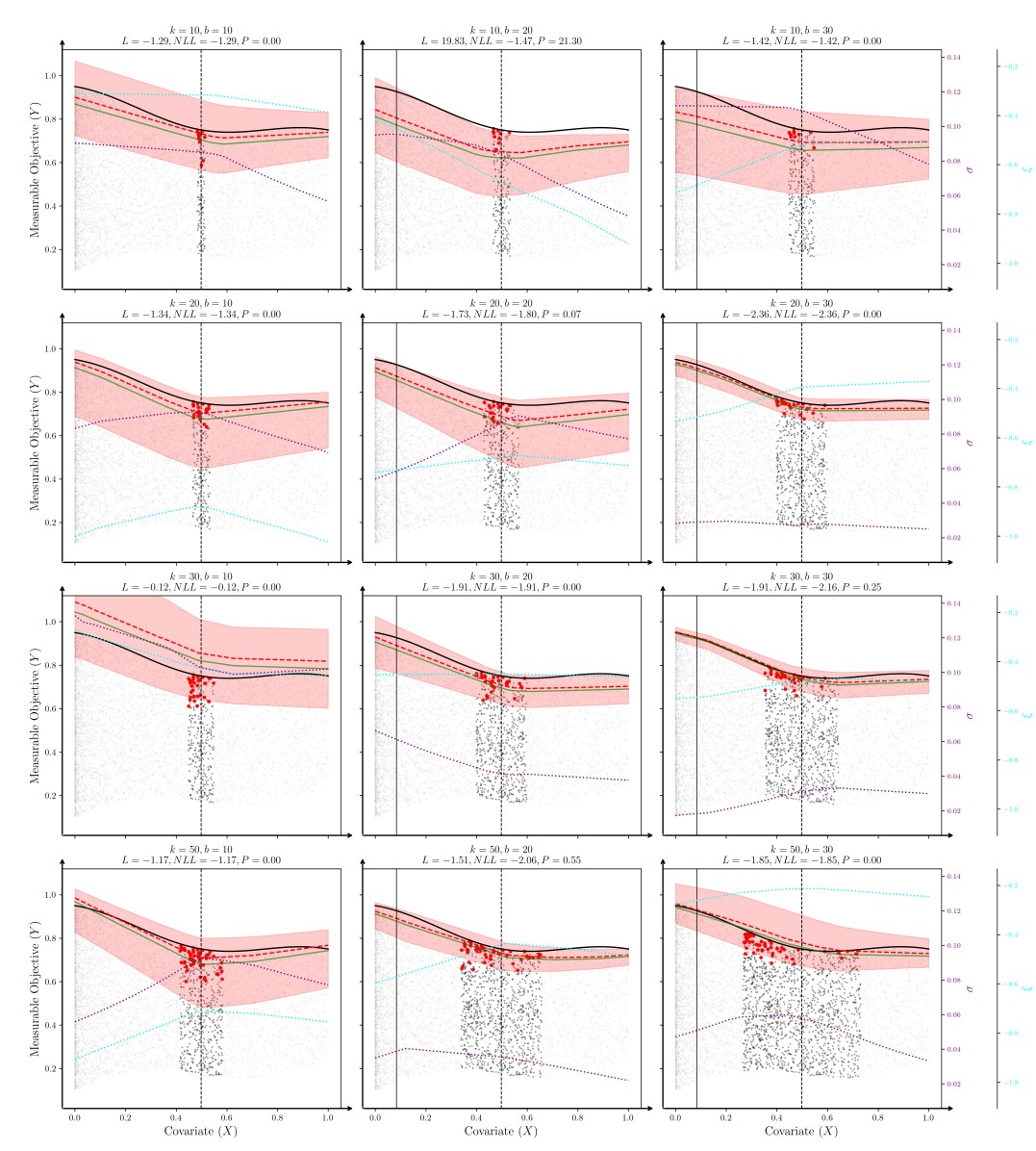

Figure 10: Regularization = 0.

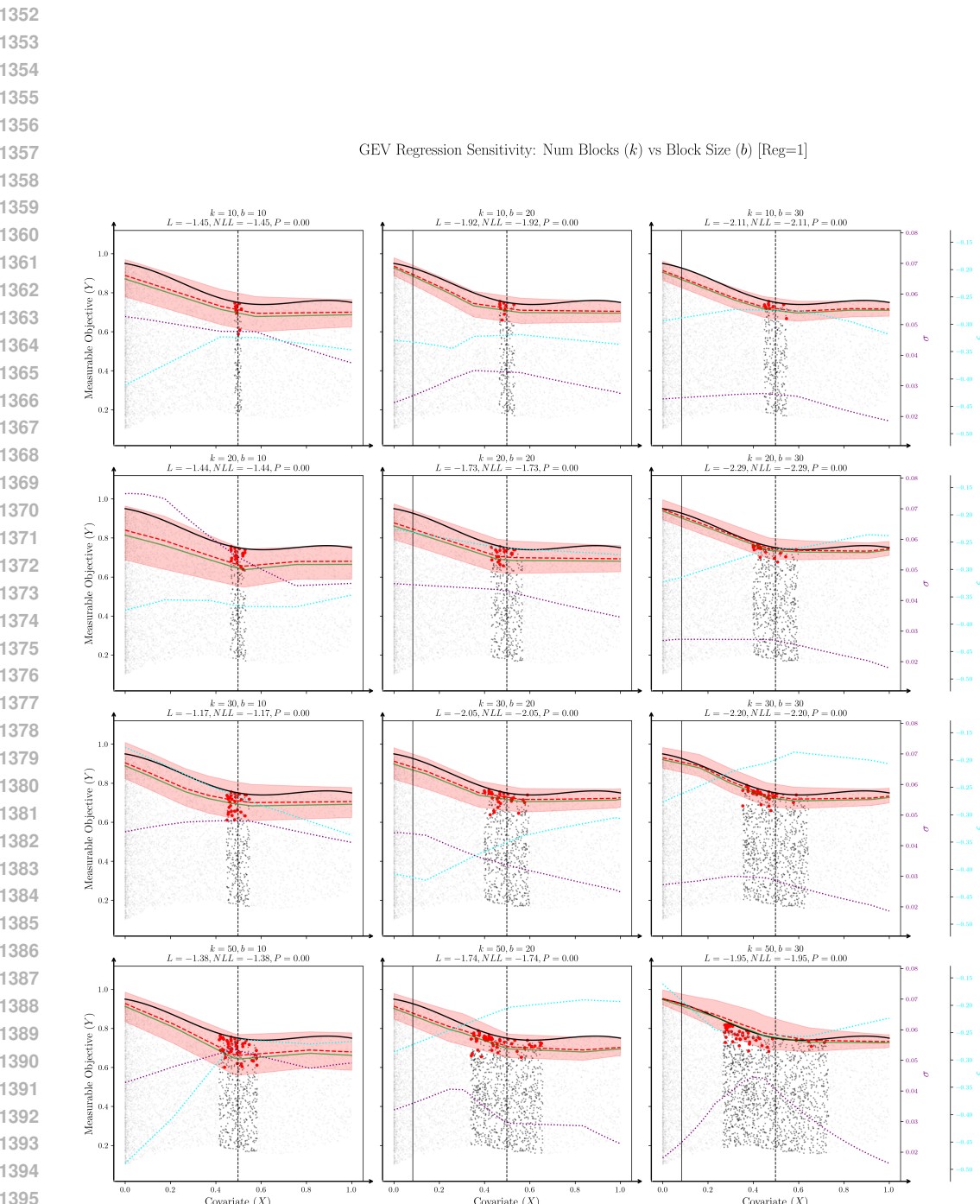

Figure 11: Regularization = 1.

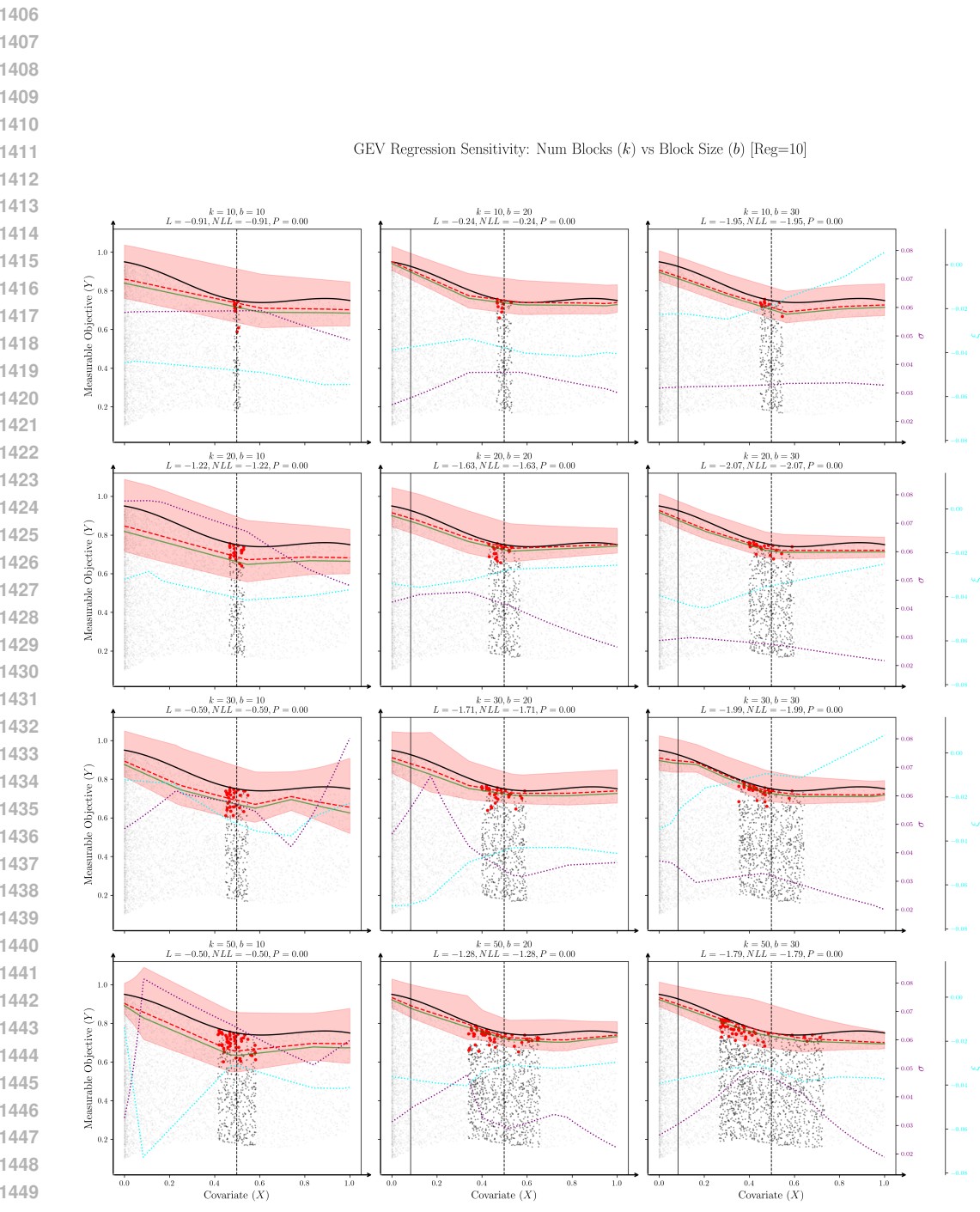

Figure 12: Regularization = 10.

## K  Experiment Details

### K.1  Baseline Methods

**Doubly-Robust Policy Learning Methods**  Conventional methods for policy learning take 2-step approaches: (1) Estimate the counterfactual outcomes for each individual-action pair; and (2) Learn a policy that gives the counterfactual-outcome-maximizing actions for given individuals. Notably among them is `DRPolicyForest`, a method derived from the doubly-robust policy trees (Athey & Wager, 2021), which estimates the probability that an action $a$ is optimal for a unit characterized by covariates $x$. The original `DRPolicyForest` maps covariates to their most-probable optimal actions; since we aim to map covariates to a distribution of actions, we modify the mapping by introducing randomness: given covariates $x$, instead of always selecting the most-probable action, we sample actions according to their probabilities of being the optimal action.

**Black-Box Optimization (BBO) Methods**  BBO methods are a class of general-purpose methods aiming at finding the input $x^*$ that maximizes an unknown function $f$ by observing a set of $(x, f(x))$ pairs. Notable BBO methods include the Gaussian Process Upper Confidence Bound (`GP-UCB`) algorithm (Srinivas et al., 2010) and denoising diffusion optimization models (`DDOM`) (Krishnamoorthy et al., 2023). We adapt the `GP-UCB` algorithm to our setting as we consider the *consistency* assumption and thereby view the covariate ($\mathcal{X}$)-action ($\mathcal{A}$) joint space as the input space to the data generating process of the counterfactual outcome $y : \mathcal{X} \times \mathcal{A} \mapsto \mathbb{R}$. We adapt the `DDOM` method by joining the covariate space $\mathcal{X}$ with the outcome space $\mathbb{R}$ as the new condition space.

**`GenNOP` Ablations**  To validate the components of `GenNOP` empirically, we conduct the following ablation studies:

- Set the numerator of the learning objective weight, $g_\epsilon(y, x)$, to 1. This ablation effectively removes the "filter", thereby allowing the generative model to be trained on suboptimal actions.
- Set the denominator of the learning objective weight, $p(a|x)$, to 1. This ablation removes inverse probability weighting, thereby allowing selection biases to continue to exist in the training data.

### K.2  Evaluation Metrics

As compared to a ground-truth $\epsilon$-optimal policy, we would like the actions sampled from the generative policy $\pi_\theta$ to be both precise and comprehensive. To this end, we define the sample-based precision and recall metrics as follows:

- Precision: Expected fraction of the $m$ generated treatments that are in $\Omega_\epsilon^*(x)$. Generative policies with high precision is desirable because including more suboptimal actions (*i.e.*, actions with counterfactual measurable objective value below $y^*(x) - \epsilon$) increases the risk that the human decision-maker selects an action leading to a lower $V$ (overall utility) than the $Y$ (measurable objective)-maximizing action.
- Recall: Expected fraction of $\Omega_\epsilon^*(x)$ that have at least one corresponding generated action. The purpose of sacrificing $Y$ is to afford more opportunity to maximizing the overall utility $V$. To this end, higher diversity among the generated actions is desirable as higher $U$ (human-centered objective) values are more likely to appear, thereby leading to $V$ values higher than that of the $Y$-maximizing action.
- Fréchet inception distance (FID), which measures the difference between the generated action distribution and the ground-truth action distribution in the semi-synthetic dataset. See Appendix K.3 for details.

### K.3  Synthetic Experiment Details

**Fully-Synthetic Dataset**  We created a synthetic dataset with 1-dimensional $\mathcal{A}, \mathcal{X}$ with the following data generating process:

$$X_i \sim \text{Unif}(0,1),$$
$$\alpha_i = 10X_i + 1,$$
$$\beta_i = 12 - \alpha_i,$$
$$A_i \sim \text{Beta}(\alpha_i, \beta_i),$$
$$Y_i = \exp(-50(A_i - X_i)^2)\sin(9\pi A_i)$$

Using the above relationship, we sample $10,000$ units. Under this setting, the distribution of $\pi_\epsilon$ is multi-modal for any $x \in X$.

**Semi-Synthetic Dataset**  We created a semi-synthetic dataset derived from the `Fashion-MNIST` dataset. In essence, our task is to learn from the fashion preferences of different demographic profiles so as to provide them targeted recommendations of images of fashion items. Specifically, we sample $100,000$ individuals with 2-dimensional demographic profiles: age and gender. We assume there are 6 archetypes of demographic profiles based on age and gender, each having a mapping from the 10 classes of fashion items to $Y$ values. All items in the same class have the same $Y$ value for the same archetype. The probability that an individual of some archetype choosing one of the 10 classes of fashion items is a function of the $Y$ values of their archetype. This is in part to ensure the *overlap* assumption is met. After the individual chooses the class, they will choose one of the $5,000$ images with equal probability from the *training* set of the `Fashion-MNIST` dataset as their self-selected treatment. For each action an individual takes, we add a small Gaussian noise with mean $0$ to the $Y$ value mapped from the archetype of that individual as the observed $Y$ value.

We trained a classifier that maps an image to one of the 10 classes of fashion items, which is used to calculate the precision and recall metrics. In addition to the precision and recall metrics, we also report the FID: For each profile, we randomly draw 5 classes with replacement from its $\epsilon$-optimal classes with equal probability and then draw one of the $1,000$ images from the `Fashion-MNIST` *test* set for each class we draw also with equal probability. Representing the generated and the ground-truth distributions of $\epsilon$-optimal policies empirically both using the 5 independently sampled actions, we calculate the FID between the two empirical distributions.

**End-to-End Dataset**  We created a synthetic dataset with 1-dimensional $\mathcal{A}, \mathcal{X}$ with the following data generating process:

$$X_i \sim U(0.1, 0.9)$$
$$y_{\text{width},i} = 50 + 25\sin(2\pi X_i)$$
$$u_{\text{width},i} = 25 + 10\cos(3\pi X_i)$$
$$y_{\text{center},i} = X_i$$
$$u_{\text{center},i} = X_i + 0.4 + 0.15\sin(4\pi X_i)$$
$$Y_i(a) = \exp(-y_{\text{width},i}(a - y_{\text{center},i})^2) + 0.3\exp(-40(a - (y_{\text{center},i} - 0.25))^2)$$
$$U_i(a) = \exp(-u_{\text{width},i}(a - u_{\text{center},i})^2) + 0.2\exp(-30(a - (u_{\text{center},i} + 0.2))^2)$$
$$V_i(a) = Y_i(a)^{0.6} \cdot U_i(a)^{0.4}$$

## K.4    REAL EXPERIMENT DETAILS

Human experts solve utility-maximization problems involving measurable and human-centered objectives implicitly in their decision-making. Without access to ground-truth $\epsilon$-optimal policies, we can nonetheless treat the decisions made by expert-level human decision-makers as a proxy for the ground truth. To this end, we study medications prescribed to patients admitted to intensive care units (ICUs) to showcase the alignment between the recommendations from `GenNOP` and the prescriptions given by critical care practitioners. The goal of the decision problem facing the latter can be characterized as significant in both measurable and human-centered objectives: critical care practitioners are tasked with stabilizing patients as their immediate goal, while they must take a holistic approach towards caregiving with their expertise and intuition. Additionally, there are vast pools of

past patient records available electronically for us to apply `GenNOP`. Among them is Medical Information Mart for Intensive Care (MIMIC)-IV (Johnson et al., 2023), a large de-identified dataset available for credentialed access. From its `icu` module, we extract 2 datasets explained as follows:

**mimic-icu-sepsis**   Sepsis is a life-threatening condition often treated in ICUs and is a major cause of mortality worldwide. It requires timely and appropriate antibiotic therapy to improve outcomes. The challenge lies in selecting the correct antibiotic regimen due to the diverse range of potential pathogens, including bacteria, viruses, and fungi (Gauer et al., 2020). Recent advancements in machine learning have shown promise in predicting sepsis outcomes and optimizing treatment strategies (Raghu et al., 2017; Moor et al., 2021). Prompt and effective intervention, supported by machine learning models and clinical tools, can significantly enhance patient recovery and reduce the burden of sepsis on healthcare systems.

In our study, we selected $2,783$ patients whose diagnoses fell under the sepsis category using a set of International Classification of Diseases Version 9/10 (ICD-9/10) codes for sepsis-related diseases, who were admitted to the ICU, and who were eventually discharged from the hospital having survived, by joining tables and filtering columns. We regard individual age, sex, and SOFA score as covariates ($X$). SOFA (Sequential Organ Failure Assessment) scores are a valuable tool in critical care for assessing organ dysfunction and predicting patient outcomes in sepsis patients (Raith et al., 2017). We consider the dosages of $4$ commonly prescribed antibiotics (Vancomycin, Meropenem, Piperacillin/Tazobactam, and Azithromycin) in the patients' initial prescription as the treatments. The total length of stay (LOS) in the hospital is a significant treatment indicator for sepsis patients, as it reflects the severity of the illness, the effectiveness of the treatment, and the patient's response to therapy. Prolonged LOS is often associated with increased hospital costs, higher mortality rates, and a greater likelihood of long-term complications (PL et al., 2024). We thus consider negative hospital LOS (to conform to the maximization problem of `GenNOP`) as the measurable objective ($Y$).

**mimic-icu-cardio**   Cardiovascular diseases are a leading cause of morbidity and mortality worldwide and often necessitate intensive care for appropriate management. In our study, we selected $4,219$ patients whose diagnoses fell under the cardiovascular disease category, who were admitted to the ICU, and who were eventually discharged from the hospital having survived, by joining tables and filtering columns. We consider individual age, sex, and lactate level as covariates. The treatments of interest are the dosages of four commonly administered cardiovascular medications: norepinephrine (a vasopressor), heparin (an anticoagulant), furosemide (a diuretic), and metoprolol (a beta-blocker). As with the sepsis dataset, the total hospital length of stay (LOS) is used as the outcome variable, reflecting both the severity of cardiac conditions and response to treatment.

**Evaluation**   Since we have no access to the counterfactual outcomes under actions other than those recorded in the datasets, we cannot calculate those metrics reported for synthetic and semi-synthetic datasets which compare generated actions with ground-truth $\epsilon$-optimal actions. Instead, we regard the actions taken by critical care practitioners filtered by the fitted max-stable process as a proxy for the ground truth. We reserve a small range of covariates solely for this purpose and exclude all actions within that range from the training set to prevent leakage that inflates evaluation metrics.

We used principal component analysis (PCA) to reduce the action-space dimensionalities (*i.e.*, numbers of medications considered) for both real datasets to $2$. For each dataset, we generate $\epsilon$-optimal action samples for the reserved range of covariates. We use two methods to generate action samples: (1) `GenNOP` and (2) perturbation of optimal actions: we select the action with the highest observed $Y$ value (*i.e.*, shortest hospital LOS) and sample from the multivariate Gaussian distribution with the selected action as the mean and the covariance matrix of all non-reserved actions. To represent the densities of the generative distributions, we apply Gaussian kernel density estimation (KDE) to the generated action samples.

If we assume the actions taken by human experts, provided that they are $\epsilon$-optimal, follow the distribution given by the $\epsilon$-optimal policy, we can assess the performance of any generative distribution by comparing it with the $\epsilon$-optimal actions taken by human experts. We trained a separate point-estimate model for $y^*(x)$ which we used to select the $\epsilon$-optimal actions taken by human experts to avoid reusing part of the method being evaluated (*i.e.*, max-stable process regression in `GenNOP`).

## L    FURTHER DISCUSSIONS ABOUT DECISION-MAKER PERCEPTION PREFERENCES AND CAPABILITIES

As expected, the ground-truth (a) $\epsilon$-optimal policy with the best $\epsilon$ achieves zero regret at $(\lambda_Y, \lambda_U, \lambda_V) = (0, 1, 0)$ and trivially at $(\lambda_Y, \lambda_U, \lambda_V) = (0, 0, 1)$. Conversely, reducing the hybrid system to the machine-only system at $(\lambda_Y, \lambda_U, \lambda_V) = (1, 0, 0)$ yields the worst possible regret under $\delta = 0$. Zero regret is also achievable as long as $\lambda_Y$ is below a threshold determined by the ratio between $\lambda_U$ and $\lambda_V$. Notice that the threshold is higher when the ratio is higher. This indicates that decision-makers implementing `GenNOP` are better off *not* considering the overall objective and focusing on the human-centered objective instead when they have moderate levels of perception preference for the measurable objective that cannot be reduced in practice. We observe that (b) `GenNOP` can no longer achieve zero regret due to its non-zero–albeit diminishing–densities in the region where $Y_a < y^* - \epsilon$. Depending on the geometry of the true $\epsilon$-optimal region in $\mathcal{A}$, generative models underlying `GenNOP` may make varying levels of errors as they implicitly interpolate and extrapolate. This problem can be ameliorated by introducing classification model ($\mathcal{A} \mapsto \{0, 1\}$) acting as a post-generation filter that further diminishes the densities in the region where $Y_a < y^* - \epsilon$. `GenNOP` nevertheless enjoys low regret under all but the perception preferences closest to the $Y$-vertex. We notice that the regret at the $U$-vertex is slightly higher than that slightly farther away from the $U$-vertex due to the generative model errors. Fortunately, real-world decision-makers can rarely operate at the $U$-vertex even when instructed to do so; they likely operate with the ideal perception preferences when generative model errors are considered. Comparing (b) with (f), we validate the superior performance of `GenNOP` over the baseline method `DDOM` as the undesirable region close to the $U$-vertex of `GenNOP` is much smaller.

The performance of `GenNOP` is dependent on the choice of $\epsilon$. As $\epsilon \to 0$, `GenNOP` is reduced to an optimization algorithm; and as $\epsilon \to \infty$, `GenNOP` is reduced to an unconditional generative model trained from all observational actions. Since decision-makers do not have access to the value of the best $\epsilon \approx 0.2$, we repeat the analysis for a small $\epsilon = 0.05$ and a large $\epsilon = 0.5$ in (d) and (e) to probe the behavior of `GenNOP` under misspecified $\epsilon$'s. Zero regret is not attainable in either case. For the small $\epsilon$, low-to-moderate regret is attainable outside the immediate neighborhood of the $Y$-vertex, compatible with most decision-makers at the expense of higher attainable minimum regret. The large $\epsilon$ case resembles the `DDOM` case (which is not dependent on $\epsilon$) but with higher minimum regret. Consequently, we recommend decision-makers implementing `GenNOP` to choose the value of $\epsilon$ conservatively.

Decision-makers are limited by their perception *capabilities* regardless of their perception *preferences*. A decision-maker may attempt at executing a perception preference faithfully and consistently, yet they may not choose the action maximizing $Q_a(\lambda_Y, \lambda_U, \lambda_V)$ as they have access to only a noisy version of $\{Q_a\}_{a \sim \pi}$. We set $\delta = 0.2$ in (c) so that the perceived quality of each action is multiplied by an independent factor uniformly drawn from $[0.8, 1.2]$; the order statistics, especially the maximum, are likely perturbed by this random factor. We observe that `GenNOP` is reasonably robust under this perturbation, as the perturbed case largely resembles the unperturbed case with only a slight increase in minimum regret and in the size of the undesirable region near the $Y$-vertex.

# M    ADDITIONAL EXPERIMENTAL RESULTS

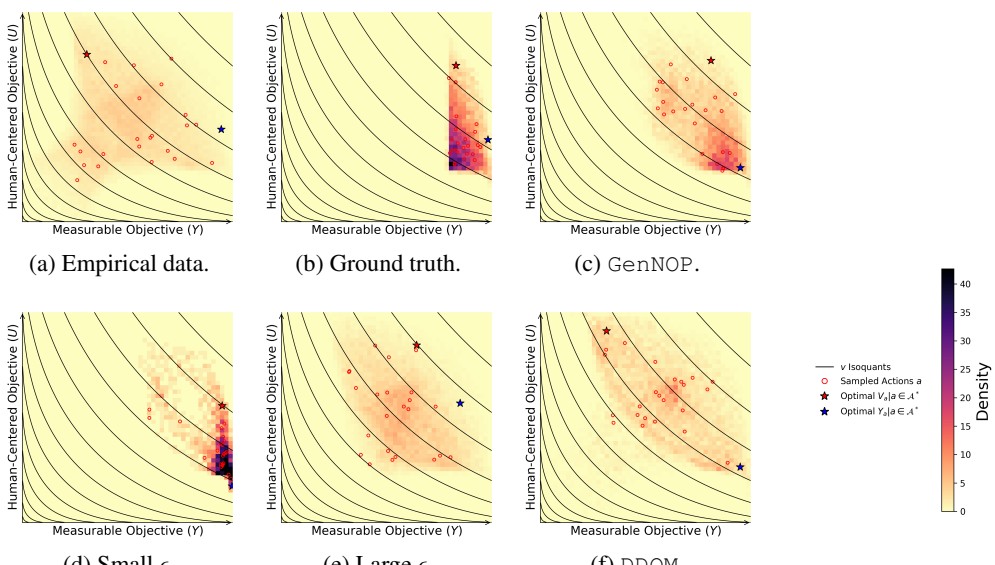

Figure 13: Action Distributions on the $Y - U$ Plane.

