# OpenReview forum: "Optimizing the Ineffable: Generative Policy Learning for Human-Centered Decision-Making"
_ICLR.cc/2026/Conference — Submitted to ICLR 2026_

### Official Review · Reviewer_ra3X · 2025-10-30

**Soundness:** 3
**Presentation:** 1
**Contribution:** 2
**Rating:** 6
**Confidence:** 2

**Summary:**

This paper proposes GenNOP, an algorithmic decision-making method that tries to align with hard-to-specify human values. They split the value function into two components, measurable and unmeasurable. Then, the algorithm optimizes for the measurable utilities, and proposes $\epsilon$-optimal candidates for the human to choose. The candidates that the human chooses are optimal in the unmeasurable sense in that they align the most with the human, and by their construction, are also value-optimal. The authors conduct both synthetic and real-world experiments to show the efficacy of their method.

**Strengths:**

* Interesting proposal to allocate roles to human and algorithm in a way that reduces human load (compared to an approach such as RLHF).
* Flexible problem formulation allows for optimality to be defined on a per-person basis, enabling satisfaction for different stakeholders.
* The experimental results support the claims in the paper.

**Weaknesses:**

* I feel the overall presentation of information in this paper can be improved. For example, Figure 1 is informative of how the whole algorithm works, but it feels like 3 figures crammed into 1. Another example, Figure 3 seems like a generic plot and hard to relate with Sec. 3.2 or the rest of the paper. See questions for more specific followups.
* Quite a bit of main paper material were discretely introduced in the Appendix. For example, the baselines (GP-UCB, DDOM, DRPolicyForest), key mathematical components ($y^*, g$), etc. As a reader, I would appreciate to be introduced to these concepts in the main paper without being required to refer to the appendix as a dependency.
* Assumption 1 plays a key role in much of the theoretical analyses, with supporting evidence from Figure 8, but does this strict concavity trend always hold? If it does not, how does this affect the $\epsilon$-optimality? I feel this should at least be mentioned more explicitly as a limitation.

Minor:
* Line 294: GEV abbrev. used before defining it (Line 1025). Same with VAE on Line 295.

**Questions:**

* What do the drawings from the zoomed-in A's on the left side mean? Should the top right plot should probably be its own figure?
* In Figure 3, where are the datapoints coming from?
* In Figure 4, what setting of m and \epsilon are you using for the left and right plots, respectively?

---

> ### Author Response · Authors · 2025-12-04
> **Responses to Questions**
>
> Thank you for your review. Please see our responses to your questions below:
>
> > What do the drawings from the zoomed-in A's on the left side mean? Should the top right plot should probably be its own figure?
>
> The drawings are stylized representations of high-dimensional actions. We removed the top right plot in favor of the new Figure 1 which now serves its intended illustrative role.
>
> > In Figure 3, where are the datapoints coming from?
>
> The left plot comes from IMDb; the right plot comes this study: van Langeveld, Astrid WB, et al. "The relationship between taste and nutrient content in commercially available foods from the United States." Food Quality and Preference 57 (2017): 1-7.
>
> > In Figure 4, what setting of m and \epsilon are you using for the left and right plots, respectively?
>
> We used $m=20$ for the left plot and the best $\epsilon$ for the right plot.

---

### Official Review · Reviewer_DEk9 · 2025-11-01

**Soundness:** 3
**Presentation:** 2
**Contribution:** 3
**Rating:** 6
**Confidence:** 4

**Summary:**

This paper proposes a new framework for algorithm-assisted decision-making for problems which they term human-centered decision-making problems: problems where the overall utility V of the decision can be separated between a quantifiable component Y and subjective, non-measurable component U that can only be evaluated by human judgement. They propose GenNOP, a method to learn policies that return not only the best performing action on Y, but a set of near-optimal actions, from which humans can chose from to factor in the decision process their evaluation of U.

**Strengths:**

- The setting of human-centered decision-making problems is interesting and well motivated
- The method proposed is sensible and principled
- Experiments on synthetic and real-world datasets show good performance

**Weaknesses:**

Major:

- The formalism in Section 2 is hard to follow. The assumptions 1-6 are only laid out in the appendix and not even all of them are mentioned in the main text. Propositions 1-3 are presented without much context or explanation as to what these results mean for the framework, and are not used or mentioned a single time in the rest of the paper. As it stands only Definition 1 is actually used in the rest of the paper. I encourage the authors to rework this section to better integrate the formalism with the discussion around it, and discuss thoroughly what each assumption and proposition mean in practice.
- In the experiments (eg first Table in section 4, Figure 6) why don’t we compare with model-free offline RL methods, for example estimating a Q function, and then picking actions uniformly within {a, ||max_a’ Q(x,a’) - Q(x,a)|| < eps} ? This seems to me like a natural baseline to compare GenNOP to so I was surprised not to see it in the experiments.
- The choice of epsilon is essential (as noted by the authors and illustrated Figure 4). Can the authors provide some discussion about how this would be done by practitioners in practice? Given the overall outcome V is never measured, does it necessarily have to be chosen “blindly” by practitioners, or can it be refined over time after collecting data from a GenNOP-like process?

Minor:

- Table of results in Section 4 is missing a caption
- Typo l.296 “an re-weighted”

**Questions:**

- In Section 3.2: I’m not familiar with the theory but as I understand it, the estimation relies on a form of continuity between the covariates X and the optimal outcomes y*(x) = max_a Y_a | X=x. That is, “close” covariates are expected to lead to similar optimal outcomes. However, in practice X can be complex and high-dimensional, eg with categorical features and sparsely sampled in the available data. Can the authors discuss these assumptions and give a sense of how robust this estimation might be in real-world datasets like the two MIMIC-IV extracts?

---

> ### Author Response · Authors · 2025-12-04
> **Response to Question**
>
> Thank you for your review. Please see our response to your question below:
>
> > In Section 3.2: I’m not familiar with the theory but as I understand it, the estimation relies on a form of continuity between the covariates X and the optimal outcomes y\*(x) = max_a Y_a | X=x. That is, “close” covariates are expected to lead to similar optimal outcomes. However, in practice X can be complex and high-dimensional, eg with categorical features and sparsely sampled in the available data. Can the authors discuss these assumptions and give a sense of how robust this estimation might be in real-world datasets like the two MIMIC-IV extracts?
>
> We did assume continuity in the covariates. The choices of the distance metric, the number of blocks ($k$), and the block size ($b$) will jointly influence this estimation. We added an ablation study showing its robustness under the synthetic settings where the Euclidean distance is used. In real-world applications, the choice of distance metrics will be dependent on prior knowledge and/or assumptions. For example, we can use pre-trained patient embedding models to project electronic medical records (EMRs) to fixed-dimensional embedding vectors, which can be used as patient covariates. By using cosine distance over this embedding space, we agree with the choices made during pre-training. We will add a robustness check over the choice of distance metrics in future revision of this paper.

---

### Official Review · Reviewer_fpSd · 2025-11-04

**Soundness:** 1
**Presentation:** 2
**Contribution:** 2
**Rating:** 2
**Confidence:** 2

**Summary:**

The paper proposes GenNOP (Generative Near-Optimal Policy learning) for human-centered decision making. The goal is to train a conditional generative model that samples actions from a target policy (\pi^_{\epsilon}(\cdot\mid x)) defined as uniform over the (\epsilon)-optimal set with respect to the measurable outcome (Y):
 [
 \Omega^{\epsilon}(x)={a:; |y^(x)-\mathbb{E}[Y^a\mid X=x]|\le \epsilon},\quad
 \pi^{\epsilon}(a\mid x)\propto \mathbf{1}{a\in\Omega^{\epsilon}(x)}.
 ]
 Training uses importance-like reweighting of an observational dataset with weights
 [
 w(x,a,y;\epsilon)=\frac{g{\epsilon}(y,x)}{p(a\mid x)},
 ]
 where (p(a\mid x)) is a generalized propensity score (GPS), and (g_{\epsilon}(y,x)=\mathbb P{y^(x) < y+\epsilon}) is estimated via a max-stable/GEV process regression on block maxima over nearest neighbors. A conditional diffusion model is then trained on the reweighted data to generate actions. Synthetic experiments illustrate regret vs. (\epsilon) and sample budget (m); real-data case studies (e.g., ICU dosing) provide qualitative visual comparisons between generated actions and filtered clinician actions.

**Strengths:**

- Timely problem & motivation. Framing human-in-the-loop selection over a diverse near-optimal set is compelling and relevant to human-centered AI.

- Engineering effort. A non-trivial pipeline (GEV nets for (g_{\epsilon}), VAE-style GPS, conditional diffusion) is implemented; synthetic experiments are thoughtfully designed.

- Synthetic results. Plots relating regret to (\epsilon) and sample budget (m) illustrate the intended trade-offs and potential practical utility when assumptions hold.

**Weaknesses:**

- The paper claims uniform sampling over (\Omega^*{\epsilon}(x)) but trains with weights (g{\epsilon}(y,x)/p(a\mid x)); no argument shows the induced generator is uniform (or even calibrated) over the (\epsilon)-optimal set. This undermines the central “diverse without historical bias” claim.

- Visual PCA/KDE alignment to filtered clinician actions (derived using the same filter) does not evaluate counterfactual utility. No off-policy evaluation, CIs, or sensitivity analyses are provided

- Guarantees are developed near the (V)-optimizer, while the method operates near the (Y)-optimizer.

**Questions:**

Can you add off-policy evaluation for continuous actions (with uncertainty), sensitivity to (\epsilon) and to the block/neighbor choices, and comparisons to learned stochastic baselines beyond Gaussian perturbations?

---

> ### Author Response · Authors · 2025-12-04
> **Response to Question**
>
> Thank you for your review. Please see our response to your question below:
>
> > Can you add off-policy evaluation for continuous actions (with uncertainty), sensitivity to (\epsilon) and to the block/neighbor choices, and comparisons to learned stochastic baselines beyond Gaussian perturbations?
>
> All of our experiments are already off-policy evaluations for continuous actions. In the synthetic experiments, we assumed data-generating processes where individuals choose actions ($A$) according to pre-defined policies which are not $\epsilon$-optimal; in the real dataset experiments, the actions were chosen by actual clinicians. Uncertainty in overall utility ($V$) is captured by the stochastic nature of our policy. In all synthetic experiments, we've already included comparisons to learned stochastic baselines (DRPolicyForest, DDOM, GP-UCB).
>
> We added an ablation study of GEV parameter estimation that shows the robustness of our method over random initializations of neural network parameters and the effects of regularization, number of blocks ($k$), and block size ($b$).

---

### Official Review · Reviewer_nPLZ · 2025-11-07

**Soundness:** 2
**Presentation:** 2
**Contribution:** 2
**Rating:** 4
**Confidence:** 4

**Summary:**

The paper presents GenOP, a generative model framework for generating sets of actions from which a decision-maker can pick its preferred one to complete a task, thus balancing normative notions and unknown user preferences. Under several theoretical assumptions, the method proposes $\epsilon$-optimal actions that stay on the Pareto frontier generated by the user’s and task’s optimal values. Lastly, under both synthetic and realistic datasets, the authors evaluate the regret-minimizing ability of their approach and the capabilities of GenOps in generating similar actions taken by physicians.

**Strengths:**

The paper tackles a very relevant problem in today’s literature. Indeed, we still lack decision support systems that enable human decision makers to achieve complementarity (e.g., the human+AI achieves superior performance than the human alone). Further, the paper also considers a relevant issue stemming from potential trade-offs between the task objective (e.g., curing effectively a patient) and some personal preferences of the decision maker.

**Weaknesses:**

- In general, the paper is very dense (and it makes it harder to address the impact of its core contribution). The proposed solution uses many different techniques (e.g., Extreme value theory, VAE, causality), but their contributions are only briefly sketched or relegated to the Appendix. Given that GenOP has many moving parts (and all of them come with their own assumptions), I believe this method could have trouble generalizing beyond the very synthetic scenario of the lab (e.g., or at least it would require a stronger experimental evaluation than the one provided).

- I believe assumptions should be stated within the main text, or at least summarized textually in layman's terms to make the reader understand the limits (and strengths) of the proposed theoretical formulation. Propositions 1,2, and 3 seem to require many assumptions (6!), but they are buried within the Appendix. I believe they need to be made more explicit in the main text.

- I do not fully understand the experimental support of the real dataset experiment (line 428). GenOP can generate a distribution of actions similar to the observed one, but I believe that is because GenOP is a generative model trained on real-world data. Thus, it is somewhat trivial that it will learn “$\epsilon$-optimal” actions close to the physician policies (furthermore, we cannot even evaluate counterfactual effects in this case). For example, it would have been nice to see the comparison between $m$ actions generated by GenOP and $m’$ actions generated by a simple VAE, to understand the benefits of the GenOP architecture.

- The paper does not mention other alternative forms of human-AI complementarity that appeared in the past years. These are important to contextualize this work within the human-AI collaboration literature. Notably, decision support systems that restrict human agency, thus preventing overreliance and leading to provable improvement in performance [1,2], and _“learning to defer”_ approaches, which learns when and how to optimally defer decisions to a human decision maker over uncertain instances  [3]. See this survey for further inputs [4].

- The code provided in the supplementary comprises only the GenOP implementation, without any code to run the experiments or the evaluations in the paper. Therefore, I believe that it is not useful in assessing the reproducibility (or to clear any of my doubts with the setup).

[1] Straitouri, E., Wang, L., Okati, N., & Rodriguez, M. G. (2023, July). Improving expert predictions with conformal prediction. In International Conference on Machine Learning (pp. 32633-32653). PMLR.

[2] De Toni, G., Okati, N., Thejaswi, S., Straitouri, E., & Rodriguez, M. (2024). Towards human-AI complementarity with prediction sets. Advances in Neural Information Processing Systems, 37, 31380-31409.

[3] Madras, D., Pitassi, T., & Zemel, R. (2018). Predict responsibly: improving fairness and accuracy by learning to defer. Advances in neural information processing systems, 31.

[4] Ruggieri, Salvatore, and Andrea Pugnana. "Things machine learning models know that they don’t know." Proceedings of the AAAI Conference on Artificial Intelligence. Vol. 39. No. 27. 2025.

**Questions:**

- Can you better state which standard causal assumptions underlie your weight function (lines 286-287)? In general, the counterfactual distribution is not identifiable from observational data, unless we have interventional data too (or unless the underlying causal generative process is invertible to some extent).

- How did you evaluate the quality of your GEV estimate (lines 315-316)? Fitting the parameters with neural networks can be brittle and give unreliable estimates depending on the data splits.

- How many runs did you compute your standard deviation on (lines 330-331)?

- Can you better describe the relevance of the “real dataset” experiment (line 428)? Can you provide a comparison between GenOP actions and the ones of a simple generative model (e.g., CVAE)?

---

> ### Author Response · Authors · 2025-12-04
> **Responses to Questions**
>
> Thank you for your review. Please see our responses to your questions below:
>
> > Can you better state which standard causal assumptions underlie your weight function (lines 286-287)? In general, the counterfactual distribution is not identifiable from observational data, unless we have interventional data too (or unless the underlying causal generative process is invertible to some extent).
>
> The standard causal assumptions underlying our weight function are (1) Consistency, (2) Unconfoundedness, and (3) Positivity. Under these assumptions, we can identify both counterfactual outcomes (which the numerator term $g_\epsilon(y, x)$ depends on) and the generalized propensity score (which is the denominator term $p(a|x)$). We added this to the paper.
>
> > How did you evaluate the quality of your GEV estimate (lines 315-316)? Fitting the parameters with neural networks can be brittle and give unreliable estimates depending on the data splits.
>
> We added an ablation study of GEV parameter estimation that shows the robustness of our method over random initializations of neural network parameters and the effects of regularization, number of blocks ($k$), and block size ($b$).
>
> > How many runs did you compute your standard deviation on (lines 330-331)?
>
> The standard deviations were computed over 10 runs. We added this to the paper.
>
> > Can you better describe the relevance of the “real dataset” experiment (line 428)?
>
> The goal of the real dataset experiments is to showcase the performance of our method when the ground-truth data-generating process is outside our control, thereby making any claim we make about the performance more robust. An ideal evaluation for these experiments would involve medical experts as raters for the generated actions under different policies in randomized controlled trials. However, since we didn't have access to a cohort of medical experts, we had to resort to clinicians whose actions we did observe in the MIMIC datasets. In essence, we assumed the actions taken in the training sets and those in the holdout sets were from different policies, and the policies of the holdout sets can be used to evaluate the learned generative policies. We acknowledge this is a strong assumption and added it as a limitation of our paper.

---

### Author Response · Authors · 2025-12-03
**Paper Revision**

Dear Reviewers,

Thank you so much for your thoughtful and constructive feedback. We have made the following revisions in response:
* Added an ablation study of GEV parameter estimation (new Table 1 on Line 358) that shows the robustness of our method over random initializations of neural network parameters and the effects of regularization, number of blocks ($k$), and block size ($b$).
* Added a figure (new Figure 1 on Line 89) that illustrates how the key quantities ($A, Y, U, V$) under our problem setup relate to one another.
* Updated a figure (new Figure 6 on Line 446) that includes more baseline methods for comparison.
* Updated a figure (new Figure 2 on Line 155) to remove a distracting element in the previous version; the new Figure 1 now serves the intended illustrative role.
* Added more human-AI complementarity literature to Related Work.
* Added limitations of our evaluation strategy.
* Corrected minor language and formatting issues throughout.

We will make the following revisions in the next version of this paper (the camera-ready version should it be accepted):
* Add more comparisons in the real dataset experiments.
* Add a robustness check over the choice of distance metrics.
* Restructure Section 2 to improve readability.
* Include code to reproduce tables and figures.

We are grateful for your input that helped us improve the paper.

---

### Meta-Review · Area_Chair_iVYC · 2026-01-15

**Summary:**

This paper studies decision-making in a setting with humans. The paper studies how to use a conditional generative model to reliably produce diverse, near-optimal stochastic properties. Reviewers had concerns that the paper was very dense and hard to read and that the emperical results do not fully support the theory in the paper. In addition, the paper fails to discuss other forms of Human-AI interaction. In addition, reviewers had some concerns about the theory results in the paper.

**Reviewer Concerns:**

Reviewers had concerns that the paper was very dense and hard to read and that the emperical results do not fully support the theory in the paper. In addition, the paper fails to discuss other forms of Human-AI interaction. In addition, reviewers had some concerns about the theory results in the paper. As the authors did not revise the paper and or give a very extensive response, these concerns are still outstanding.

**Reviewer Scores:**

I believe both negative reviewers would have maintained their score as the authors did not revise the paper or provide a substantial rebuttal.

---

### Decision · Program_Chairs · 2026-01-26

Reject